# WaveletDiff: Multilevel Wavelet Diffusion For Time Series Generation

## Abstract

Time series are ubiquitous in many applications that involve forecasting, classification and causal inference tasks, such as healthcare, finance, audio signal processing and climate sciences. Still, large, high-quality time series datasets remain scarce. Synthetic generation can address this limitation; however, current models confined either to the time or frequency domains struggle to reproduce the inherently multi-scaled structure of real-world time series. We introduce WaveletDiff, a new framework that trains diffusion models *directly on wavelet coefficients* to exploit the inherent multi-resolution structure of time series data. The model combines dedicated transformers for each decomposition level with cross-level attention mechanisms that enable selective information exchange between temporal and frequency scales through adaptive gating. It is also informed by level-specific energy constraints based on Parseval's theorem which preserve time-frequency properties throughout the diffusion process. Comprehensive tests across six real-world datasets from energy, finance, and neuroscience domains demonstrate that WaveletDiff outperforms the diffusion baselines FourierDiffusion, Diffusion-TS, and SigDiffusions on the majority of metrics, with the smallest margin over FourierDiffusion, while still achieving roughly 3× lower discriminative and Context-FID scores. Against the VAE/transformer-based MSDFormer, the results are mostly comparable, with WaveletDiff using fewer parameters and less training time on most datasets. The most revealing finding is the significant performance gap on fMRI data (in favor of MSDFormer) and EEG (in favor of WaveletDiff). This finding is explained via a careful testing/examination of the properties of wavelet coefficients for generative, as opposed to anlyses/decomposition tasks. Our code is available at https://anonymous.4open.science/r/WaveletDiff-27E9/.

## 1 Introduction

Time series data arises in diverse practical settings, including healthcare (Lee et al., 2020; van der Schaar Lab, 2019), finance (Sezer et al., 2019; Ozbayoglu et al., 2020), climate sciences (Dinku, 2019), audio processing (Mitra & Zualkernan, 2025) and engineering (Çınar et al., 2020; Lei et al., 2020). Due to various constraints, acquiring sufficiently high-quality labeled time-series datasets remains a challenge (Wang et al., 2021; Desai et al., 2025). The problem may be mitigated through synthetic time series generation, which also offers promising solutions for data augmentation (Wen et al., 2021; Le Guennec et al., 2018; Ryu et al., 2023), privacy preservation (Wang et al., 2020; Jordon et al., 2024; Nosowsky & Giordano, 2006), forecasting (Taga et al., 2025) and simulations (Nikolenko, 2021; El Emam et al., 2022).

Current time series generation methods predominantly operate either directly in the time domain or frequency domain, and come with different advantages and limitations. Time-domain approaches, including those based on GANs (Yoon et al., 2019; Pei et al., 2021; EskandariNasab et al., 2024), autoregressive (Salinas et al., 2019) and diffusion models (Lim et al., 2023; Narasimhan et al., 2024; Sikder et al., 2024) are well-suited for modeling local temporal patterns, but struggle with long-term dependencies and preservation of important spectral characteristics. To address time-domain induced limitations, recent approaches have increasingly leveraged frequency-domain analysis, often along with temporal modeling (Tian et al., 2020; Chi et al., 2024; Crabbé et al., 2024; Huang et al., 2024). These methods are of relevance since many real-world time

series tend to exhibit higher localization in the frequency rather than the time domain. Representative methods include FourierFlow (Alaa et al., 2021), which applies normalizing flows to Fourier representations, DiffusionTS (Yuan & Qiao, 2024), which combines Fourier decompositions with diffusion models, and various frequency-enhanced transformers (Zhou et al., 2022a;b; Xu et al., 2024; Yi et al., 2023) that use both spectral and temporal analyses. However, these approaches typically process time and frequency domain information either in a separate manner or impose trade-offs between temporal resolution and spectral coherence. They are also not able to simultaneously capture both local and global time and spectral patterns, which is crucial for synthesizing realistic time series.

Wavelet transforms represent a natural approach to address the above issue by creating a multi-resolution representation that simultaneously captures both temporal and spectral information (Mallat, 1989; Cohen, 2001). Unlike the Fourier transform, which captures global frequency properties, wavelets maintain temporal localization while also providing useful decompositions into multiple frequency bands (Rioul & Flandrin, 1992; Daubechies, 1988). This results in a highly versatile time-frequency hierarchical representation (Mallat, 1989). As a result, wavelet-based analyses have been used with success for various signal processing applications including speech recognition, financial trends analysis, image processing, and biomedical signal analysis (Daubechies, 1992; Vetterli & Kovačević, 1995; Burrus et al., 1998). Despite these results, a handful of known wavelet-based approaches for time series generation have failed to provide improvements over Fourier-based methods (Takahashi & Mizuno, 2024; Kazemi & Meidani, 2022; Zhao et al., 2018). This may be attributed to the fact that, almost exclusively, the methods treat wavelet coefficients as image structures and then follow up by applying standard image generation techniques such as convolutional neural networks or image-based diffusion models. While potentially useful for data-poor applications and highly specialized time series, indirect time series → wavelet → image conversion methods in general suffer from pattern distortions caused by noninvertible image features.

A more adequate approach based on wavelet decompositions is to run diffusion models directly in the wavelet domain, which is a new direction proposed in this work. For diffusion, methods such as denoising diffusion probabilistic models (DDPMs) (Ho et al., 2020) which have demonstrated remarkable success in image (Dhariwal & Nichol, 2021), audio (Kong et al., 2021), and text generation (Austin et al., 2021), are considered state-of-the-art for time-series generation. However, these diffusion models are tailor-made for highly specific time-series formats (e.g., audio or financial data), and may not be suitable for other modalities. This motivates us to implement a new wavelet-space diffusion model, termed WaveletDiff, which is universally applicable as it inherently respects different multi-level structures. Unlike frequency-domain approaches, WaveletDiff also captures temporal patterns at different scales simultaneously. Our key innovations lie in running forward diffusion processes for each wavelet level individually and in parallel, following fine-tuned exponential noise mechanisms and using dedicated level-transformer denoising networks combined with a cross-level attention mechanism that enables information exchange between different decomposition levels. This design preserves the hierarchical nature of wavelet representations while allowing the model to learn complex inter-scale dependencies crucial for realistic time series generation.

Switching from time to frequency domain diffusion models has led to strong empirical gains and established transform-domain data generation as a compelling direction. WaveletDiff offers significant improvements over the only other direct-transform approach (Fourier), and these gains come from design choices that exploit the multiresolutional structure of wavelet decompositions — per-level processing, time–frequency coupling and cross-level attention. As a result, wavelet-based diffusion represents a new generative modeling method with potentially many additional applications.

Our technical contributions include:
**1.** A diffusion framework that operates directly in the wavelet domain and tunes the noise addition process to the approximation and detail levels, identifies the most suitable choice of mother wavelet for different time series and uses level-specific loss functions and transformers.
**2.** A cross-scale attention mechanism that enables information flow between different temporal scales while preserving their individual properties.
**3.** A wavelet-aware loss weighting mechanism that prevents some levels from dominating the training objective through level-specific balancing strategies.
**4.** A new evaluation metric based on the Dynamic Time-Warping distance and extensive comparative

analysis of both short and long time series datasets, including ETTh1, ETTh2, Stocks, Exchange Rate, fMRI, EEG (Zhou et al., 2021; Lai et al., 2018; Roesler, 2013). The results show significant performance gains of WaveletDiff compared to Fourier and time-based methods, which are roughly three-fold on average for discriminative and Context-FID scores.

**5.** Our results also reveal for the first time that signal features amenable for wavelet analysis (e.g., as is the case for fRMI data) may hurt generative models that operate in the wavelet domain.

**6.** The first empirical evaluation of reproducibility of diffusion models for time series, akin to recent efforts reported for images (Zhang et al., 2024b; Li et al., 2024; Kadkhodaie et al., 2024), as well as an added verification of WaveletDiff being robust in the context of irregularly sampled time series generation.

## 2 Related Work

### 2.1 Time Series Generation with Diffusion Models

Early generative AI methods for time series focused on conditional generation tasks such as forecasting and imputation (Rasul et al., 2021; Tashiro et al., 2021; Li et al., 2022; Yang et al., 2024; Pei et al., 2025; Hou et al., 2025; Shen et al., 2024; Fadlon et al., 2025), while recent approaches target unconditional time series generation (Shen & Kwok, 2023; Barancikova et al., 2025; Li et al., 2025b; Suh et al., 2025; Naiman et al., 2024c). The above methods employ various architectural choices including RNNs, transformers, and specialized denoising networks capable of handling sequential temporal data (Kong et al., 2021; Naiman et al., 2024a), and almost exclusively operate in the time domain. This limits their ability to capture *both* global and local, potentially varying, spectral properties.

### 2.2 Frequency and Other Transform Domain Approaches for Time Series

Recent works have demonstrated that real-world time series are more localized in the frequency domain, making spectral diffusion more effective than time diffusion (Crabbé et al., 2024). Various frequency-based approaches include lightweight models using complex-valued operations (Xu et al., 2024), frequency-enhanced transformers combining discrete Fourier transform (DFT) with attention mechanisms (Zhou et al., 2022b;a), and specialized MLP architectures for frequency learning (Yi et al., 2023). Additional methods incorporate spectral filtering (Zhang et al., 2024a), multi-resolution frequency analysis (Wang et al., 2024), and normalizing flows within the Fourier domain (Alaa et al., 2021). Additional unconditional generation approaches include interpretable diffusion models that combine *trend and seasonality* components (Yuan & Qiao, 2024) and latent diffusion models that operate in compressed latent spaces for more efficient generation (Qian et al., 2024). These methods often require separate processing pipelines for temporal and frequency components, limiting their ability to simultaneously capture multi-scale temporal-spectral relationships.

Several other lines of work perform diffusion in transformed domains to expose spectral or multi-scale structure that is difficult to model in the time domain alone. Frequency-domain approaches leverage Fourier or frequency-aware representations to better control global spectral properties and periodic patterns (Crabbé et al., 2024; Luo et al., 2025; Naiman et al., 2024a). While transform-domain diffusion for time series remains relatively rare, diffusion models have been explored in other transformed domains, primarily for audio and vision. Spectrogram-domain diffusion leverages short-time Fourier transform (STFT) representations to capture localized time–frequency structure for audio and music generation (Zhu et al., 2023; Vanukuri et al., 2025; Hawthorne et al., 2022). Scattering-transform–based approaches use wavelet scattering features as conditioning to guide diffusion, benefiting from their stability to deformations and multi-scale sensitivity (Mao et al., 2025; Dong et al., 2025; Gama et al., 2018). Discrete cosine transform (DCT)–based diffusion performs generation directly in the DCT space, exploiting compact and decorrelated representations for efficiency and downstream tasks (Ning et al., 2024; Zhong et al., 2025).

### 2.3 Wavelet-Based Time Series Modeling

Wavelet transforms provide multi-resolution time-frequency representation capabilities (Mallat, 1989; Daubechies, 1992; Addison, 2017) and have been extensively used in time series analysis (Percival & Walden, 2000; Sang, 2013; Patrik et al., 2015; Han et al., 2019). For generative modeling, wavelets have seen more

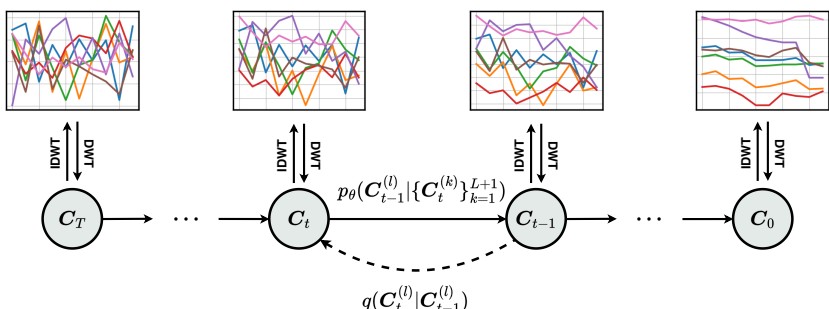

Figure 1: Direct wavelet coefficient diffusion, where the forward process proceeds independently at each decomposition level, while the reverse process integrates information across all levels for joint denoising.

limited applications, despite showing promise in terms of direct coefficient processing (Phung et al., 2022; Hu et al., 2023; Guth et al., 2022). For time series forecasting, wavelets have been employed to enhance traditional forecasting models through their multi-resolution time-frequency analysis capabilities (Zhou et al., 2025; Sasal et al., 2022; Arabi et al., 2024; Schlüter & Deuschle, 2010; Li et al., 2025a; Zhang & Wan, 2017). For generation tasks, existing methods predominantly convert wavelet coefficients to image representations for processing with standard computer vision techniques (Takahashi & Mizuno, 2024; Kazemi & Meidani, 2022). However, it is not clear that these indirect approaches fully exploit the hierarchical, global/local multi-scale structure of wavelet decompositions, due to the use of patches and/or potential incompatibilities between the image representations and actual level-dependent temporal and spectral characteristics. As a result, their generative capabilities are modest for long time series with long-range contexts.

## 3 Methodology

### 3.1 Wavelet Representations of Time Series

A multivariate time series dataset $\mathbf{X} \in \mathbb{R}^{N \times T \times D}$ with $N$ samples, $T$ timesteps and $D$ features (e.g., opening price, closing price, high/low, volume for financial data) comprises time series of the form $\mathbf{x}^{(i)} = [\mathbf{x}_0^{(i)}, \mathbf{x}_1^{(i)}, \ldots, \mathbf{x}_{T-1}^{(i)}] \in \mathbb{R}^{T \times D}$, where $i \in [1, N]$. The Discrete Wavelet Transform (DWT) decomposes each time series through a cascade of high-pass and low-pass filtering operations followed by downsampling. The decomposition utilizes a scaling function $\phi(t)$ and its associated *mother wavelet* $\psi(t)$. The mother wavelet is characterized by its order $p$, which ensures that the wavelet is orthogonal to all polynomials of degree less than $p$ (hence, $p$ determines the number of vanishing moments). Higher-order wavelets provide better frequency localization but require longer filters. The filter length $F$ represents the number of nonzero coefficients in the discrete filters, which depends on the wavelet family and order $p$ (e.g., $F = 2p$ for Daubechies wavelets). More details are available in Appendix A. These functions satisfy the two-scale relations:

$$\psi(t) = \sqrt{2} \sum_{k=0}^{F-1} g_k \, \phi(2t - k), \qquad \phi(t) = \sqrt{2} \sum_{k=0}^{F-1} h_k \, \phi(2t - k). \tag{1}$$

where $\{g_k\}_{k=0}^{F-1}$ and $\{h_k\}_{k=0}^{F-1}$ are the high-pass and low-pass filter coefficients, respectively, with the relationship $g_k = (-1)^k h_{F-1-k}$ ensuring orthogonality. The DWT performs recursive decomposition over $L$ levels. Starting with the approximation coefficients $\boldsymbol{A}^{(0)} = \mathbf{X}$, at each level $l \in [1, L]$, we apply high-pass and low-pass filters followed by temporal-dimension downsampling:

$$
\begin{aligned}
\boldsymbol{C}_{:,m,:}^{(l)} &= \sum_k g_k \, \boldsymbol{A}_{:,2m-k,:}^{(l-1)} \ \text{(detail coeff.)}, \\
\boldsymbol{A}_{:,m,:}^{(l)} &= \sum_k h_k \, \boldsymbol{A}_{:,2m-k,:}^{(l-1)} \ \text{(approximate coeff.)},
\end{aligned}
\tag{2}
$$

where $m$ indexes the downsampled time dimension and the operation is applied independently across all $N$ samples and $D$ features. Boundary effects are handled using symmetric extension, where the signal is mirrored at the tails to ensure sufficient coefficients for filtering operations. This decomposition yields the wavelet coefficient representation:

$$\text{DWT}(\mathbf{X}) = \{\boldsymbol{C}^{(1)}, \ldots, \boldsymbol{C}^{(L)}, \boldsymbol{A}^{(L)}\}, \tag{3}$$

where $\boldsymbol{C}^{(l)} \in \mathbb{R}^{N \times d_l \times D}$, for $l \in [1, L]$, are the detail coefficients and $\boldsymbol{A}^{(L)} \in \mathbb{R}^{N \times d_L \times D}$ are the approximation coefficients. For consistency with diffusion notation, we write $\boldsymbol{C}^{(L+1)} = \boldsymbol{A}^{(L)}$.

The wavelet order $p$ is chosen based on the sequence length to ensure sufficient coefficients at each level, with longer sequences accommodating higher-order wavelets for better frequency localization. The coefficient dimension at each level $l$ is calculated recursively as:

$$d_l = \lfloor \frac{d_{l-1} + F - 1}{2} \rfloor, \quad l = 1, \ldots, L, \tag{4}$$

where $d_0 = T$ is the original sequence length. This formula accounts for the filter overlap (requiring $F - 1$ additional boundary coefficients) and dyadic downsampling (division by 2) inherent to the wavelet decomposition process. The number of decomposition levels $L$ is determined based on the sequence length $T$ to ensure sufficient coefficients are available at each level while maintaining meaningful frequency separation, and is set to $L = \max\left(3, \min\left(7, \lfloor \log_2\left(\frac{T}{F-1}\right) \rfloor\right)\right)$ in practice.

To reconstruct time series from diffusion-generated wavelet coefficients $\{\hat{\boldsymbol{C}}^{(1)}, \ldots, \hat{\boldsymbol{C}}^{(L)}, \hat{\boldsymbol{C}}^{(L+1)}\}$, we apply the Inverse Discrete Wavelet Transform (IDWT). The reconstruction proceeds from the coarsest level to the finest level. Starting with $\hat{\boldsymbol{A}}^{(L)} = \hat{\boldsymbol{C}}^{(L+1)}$, for each $l = L, \ldots, 1$, we compute:

$$\hat{\boldsymbol{A}}^{(l-1)}_{:,m,:} = \sum_k \tilde{h}_{m-2k} \hat{\boldsymbol{A}}^{(l)}_{:,k,:} + \sum_k \tilde{g}_{m-2k} \hat{\boldsymbol{C}}^{(l)}_{:,k,:}, \tag{5}$$

where $\tilde{h}$ and $\tilde{g}$ are the synthesis filters used for reconstruction. For orthogonal wavelets, these take the form $\tilde{h}_k = h_{-k}$ and $\tilde{g}_k = g_{-k}$, while for biorthogonal wavelets, they are independently designed dual filters that ensure perfect reconstruction. The reconstruction combines the current approximation $\hat{\boldsymbol{A}}^{(l)}$ with the detail coefficients $\hat{\boldsymbol{C}}^{(l)}$ through upsampling and filtering. The inverse transform can hence be written as:

$$\hat{\mathbf{X}} = \text{IDWT}(\{\hat{\boldsymbol{C}}^{(1)}, \ldots, \hat{\boldsymbol{C}}^{(L)}, \hat{\boldsymbol{A}}^{(L)}\}), \tag{6}$$

where $\hat{\mathbf{X}} = \hat{\boldsymbol{A}}^{(0)} \in \mathbb{R}^{N \times T \times D}$ is the reconstructed time series.

## 3.2 Wavelet-Space Diffusion Framework

We propose to run the diffusion process on wavelet coefficients using Denoising Diffusion Probabilistic Models (DDPM), as shown in Figure 1. The forward diffusion process in the wavelet domain gradually adds Gaussian noise to coefficients at all levels independently:

$$q(\boldsymbol{C}^{(l)}_t | \boldsymbol{C}^{(l)}_0) = \mathcal{N}(\boldsymbol{C}^{(l)}_t; \sqrt{\bar{\alpha}_t} \boldsymbol{C}^{(l)}_0, (1 - \bar{\alpha}_t)\mathbf{I}), \tag{7}$$

where $l = 1, \ldots, L+1$, $\alpha_t = 1 - \beta_t$ and $\bar{\alpha}_t = \prod_{s=1}^t \alpha_s$ follow standard DDPM schedules. Specifically, we adopt an exponential noise schedule $\beta_t = \beta_{start} + (\beta_{end} - \beta_{start}) \cdot (1 - e^{-\gamma \cdot t})$, where $\gamma$ is the exponential decay rate, $t \in [0, 1]$ is the normalized timestep, and $\beta_{start}$ and $\beta_{end}$ are tuneable hyperparameters. Exponential schedules are better suited for wavelet-based time series generation than cosine schedules. This may be because cosine schedules start slowly, peak mid-epoch, and then decrease, with this smooth behavior stabilizing high-dimensional data like images. In contrast, time series and their wavelet decompositions have significantly lower dimensions, benefiting from more aggressive noise injection when coupled with transformer-based denoising models used at early backwards steps.

We parameterize the reverse process using a cross-level transformer network that employs cross-attention to enable communication across levels:

$$p_\theta(\boldsymbol{C}^{(l)}_{t-1} | \{\boldsymbol{C}^{(k)}_t\}_{k=1}^{L+1}) = \mathcal{N}(\boldsymbol{C}^{(l)}_{t-1}; \boldsymbol{\mu}_\theta(\{\boldsymbol{C}^{(k)}_t\}_{k=1}^{L+1}, t), \boldsymbol{\Sigma}_\theta), \tag{8}$$

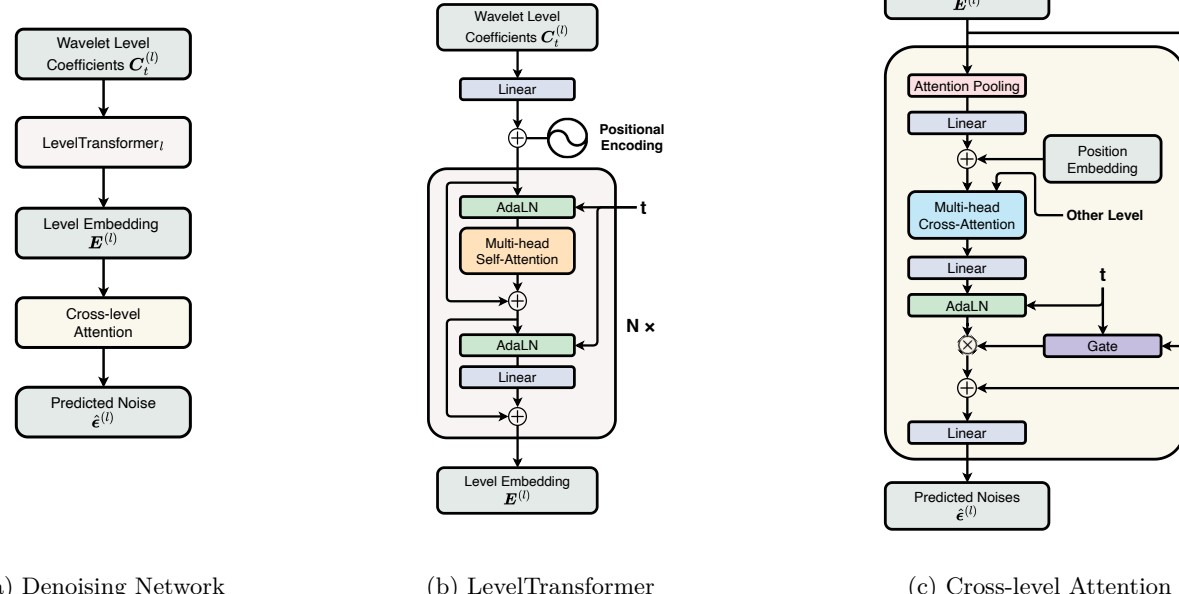

(a) Denoising Network      (b) LevelTransformer      (c) Cross-level Attention

Figure 2: The wavelet coefficients are independently processed by LevelTransformers to obtain level-specific embeddings. These embeddings are obtained through interaction across levels via a cross-level attention module based on adaptive gating mechanisms.

where $\boldsymbol{\mu}_\theta(\{\boldsymbol{C}_t^{(k)}\}_{k=1}^{L+1}, t)$ represents the predicted mean of the reverse diffusion process, parameterized by the neural network $\theta$ and conditioned on all wavelet levels and the diffusion timestep $t$, and $\boldsymbol{\Sigma}_\theta$ is the predicted covariance matrix. Following the DDPM framework, we fix $\boldsymbol{\Sigma}_\theta = \beta_t \mathbf{I}$ and train the denoising network $f_\theta$ to predict the added noise $\boldsymbol{\epsilon}$ using the mean square error (MSE) as the loss:

$$\hat{\boldsymbol{\epsilon}}^{(l)} = f_\theta(\{\boldsymbol{C}_t^{(k)}\}_{k=1}^{L+1}, t), \quad l = 1, \dots, L+1 \tag{9}$$

$$\mathcal{L}_{\text{recon}} = \mathbb{E}_{\boldsymbol{C}, t, \epsilon} \left[ \sum_{l=1}^{L+1} w_l \cdot \|\boldsymbol{\epsilon}^{(l)} - \hat{\boldsymbol{\epsilon}}^{(l)}\|^2 \right] \tag{10}$$

where $\boldsymbol{\epsilon}^{(l)}$ and $\hat{\boldsymbol{\epsilon}}^{(l)}$ are the true and predicted noise at level $l$, and $w_l$ are level-specific weights ensuring balanced contribution across scales.

To preserve data spectra for long sequence generation, we optionally introduce an energy conservation penalty based on Parseval's theorem. For orthogonal wavelet bases, we let

$$\mathcal{L}_{\text{energy}} = \mathbb{E}_{\boldsymbol{C}, t, \epsilon} \left[ \sum_{l=1}^{L+1} \left| \mathcal{E}^{(l)} - \hat{\mathcal{E}}^{(l)} \right| \right], \tag{11}$$

where $\mathcal{E}^{(l)} = \sum_{n=1}^{N} \sum_{j=1}^{d_l} \sum_{k=1}^{D} (\boldsymbol{C}_{n,j,k}^{(l)})^2$ represents the true energy at wavelet level $l$, and $\hat{\mathcal{E}}^{(l)} = \sum_{n=1}^{N} \sum_{j=1}^{d_l} \sum_{k=1}^{D} (\hat{\boldsymbol{C}}_{n,j,k}^{(l)})^2$ represents the predicted energy at wavelet level $l$. By enforcing energy preservation at each decomposition level, the constraints stabilize training and preserve the natural energy distribution across frequency scales. For biorthogonal and reverse biorthogonal wavelet bases, the corresponding energy regularization formulas are provided in Appendix B.

The overall training objective combines both the reconstruction and energy terms $\mathcal{L} = \mathcal{L}_{\text{recon}} + \lambda_{\text{energy}} \mathcal{L}_{\text{energy}}$, where $\lambda_{\text{energy}}$ denotes the weight of the energy loss term. For short sequences, the base reconstruction loss

is typically sufficient since the spectral energy drift is minimal over limited temporal horizons. The energy preservation term mostly benefits datasets with strong low-frequency trends and smooth spectral characteristics (e.g., ETTh1, Exchange Rate), while high-volatility datasets with abrupt changes (e.g., Stocks) are better reproduced through the reconstruction loss alone.

The denoising network $f_\theta$ uses dedicated transformers for each wavelet level. We adopt Adaptive Layer Normalization (AdaLN) (Peebles & Xie, 2022) as the normalization layer. For level $l$, the coefficients $\boldsymbol{C}_t^{(l)}$ are processed through a specialized transformer,

$$\mathbf{E}^{(l)} = \text{LevelTransformer}_l(\boldsymbol{C}_t^{(l)}, \mathbf{t}), \quad l = 1, \ldots, L+1, \tag{12}$$

where $\mathbf{t}$ denotes the diffusion time embedding and $\mathbf{E}^{(l)} \in \mathbb{R}^{N \times h_l \times D}$ represents the output level embeddings for level $l$, with $h_l$ denoting the embedding dimension at that level. The embeddings of each level are aggregated through attention-based pooling. Cross-level attention operates on these aggregated representations, allowing each level to adaptively incorporate contextual information from other scales through learned gating mechanisms (Figure 2).

**Discussion: Choice of the Wavelet Domain.** While diffusion models have been explored in several transform domains, including Fourier, spectrogram, scattering, and DCT representations, only Fourier-domain diffusion has been directly adopted for generic time series generation. Fourier-domain approaches provide a global frequency decomposition that is well suited for modeling stationary signals, but they lack temporal localization. As a result, Fourier coefficients entangle long- and short-term dynamics, making it difficult for diffusion models to capture transient events and regime changes that are ubiquitous in real-world time series.

On the other hand, the wavelet domain provides a multiresolution decomposition that for many types of time series separates global low-frequency structure from localized high-frequency details, naturally supporting level-wise modeling and denoising in diffusion models. Nevertheless, as will be discussed in the results section, one has to be careful to distinguish the utility of different wavelet families for signal analysis versus amenability to diffusion in the wavelet domain - in some cases, diffusion models may be outperformed by other generative modalities. The latter hinges on the statistical properties of the coefficients which in turn depend on the type of time-series to be generated.

Hence, rather than relying on qualitative arguments alone, we explicitly evaluate the effect of the transform-domain choice by comparing wavelet-domain diffusion with Fourier-domain diffusion baselines (Crabbé et al., 2024) in Section 4, with an additional focus on the choice of the wavelet family.

## 4 Experiments

### 4.1 Experimental Settings

**Benchmarks** We compare WaveletDiff to several state-of-the-art time series generation methods, including FourierDiffusion (Crabbé et al., 2024), Diffusion-TS (Yuan & Qiao, 2024), TimeGAN (Yoon et al., 2019), SigDiffusions (Barancikova et al., 2025), KoVAE (Naiman et al., 2024b), as well as MSDformer (Feng et al., 2025), which is an improved version of SDformer (Chen et al., 2024). In addition, we make use of two state-of-the-art forecasting time series foundation models (TSFMs), TimesFM (Das et al., 2024) and Chronos (Ansari et al., 2024), each comprising ∼200M parameters, to perform unconditional generation. Specifically, we sampled length-64 windows as context and autoregressively forecasted the next 24 steps to produce newly generated sequences. Although not fine-tuned for generation, these models leverage real-world inputs during generation and have large capacity. We therefore believe that considering this approach also provides a meaningful comparison.

**Datasets** We use six real-world datasets to evaluate our method, covering energy, finance, and neuroscience domains. **ETTh1** and **ETTh2** (Zhou et al., 2021) are electricity transformer datasets containing oil temperature and six power load features recorded hourly from 2016 to 2018. **Stocks** is a multivariate financial time series dataset containing historical Google stock market data with price and volume features from

2004 to 2019. **Exchange Rate** (Lai et al., 2018) contains daily exchange rates of eight countries from 1990 to 2016. **fMRI** is the NetSim dataset containing simulated BOLD time series data for evaluating network modeling methods in functional magnetic resonance imaging. **EEG** (Roesler, 2013) contains multichannel electroencephalogram recordings that measure brain electrical activity over time, offering information about neural dynamics and cognitive processes. For more details, refer to Appendix C.1.

**Metrics** We evaluate generation quality using five complementary metrics. The **discriminative score** measures similarity between real and generated samples by training a binary classifier to distinguish them (Yoon et al., 2019). The **predictive score** assesses the utility of synthetic data for forecasting real sequences using mean absolute error. **Context-Fréchet inception distance (Context-FID)** (Paul et al., 2022) quantifies distributional distance using TS2Vec (Yue et al., 2022) embeddings following (Yuan & Qiao, 2024). The **correlational score** evaluates temporal dependencies by comparing cross-correlation matrices.

Although Context-FID quantifies distributional distance, it measures distance in the embedding space of a trained TS2Vec encoder, and therefore inherits whatever structural biases that encoder may have. In particular, it is unclear whether the encoder adequately captures temporal alignment, phase shifts, or nonlinear time distortions. Dynamic Time Warping (DTW), on the other hand, is a model-free, alignment-aware similarity measure that explicitly accounts for temporal warping by finding the optimal monotone alignment between two sequences. This makes DTW sensitive to shape-based similarity in a way that embedding-based distances may not be. We therefore propose the **Dynamic Time Warping Jensen–Shannon Distance (DTW-JS distance)**, which combines DTW with Jensen–Shannon divergence to assess the differences in the distributions of DTW-based discrepancies between real and generated data. DTW aims to capture optimal temporal alignments between two time series sequences $x$ and $y$ by minimizing

$$\text{DTW}(x,y) = \min_{\pi} \sum_{(i,j) \in \pi} |x_i - y_j|, \tag{13}$$

where $\pi$ is a warping path allowing flexible temporal matching. We first create a reference set $\mathcal{M}$ by randomly sampling sequences from the union of the real $\mathcal{R}$ and generated $\mathcal{G}$ datasets, which are matched in size. For each sequence $s$ in the real dataset, we compute its mean DTW distance to all sequences in the reference set: $d_{\mathcal{R}}(s) = \frac{1}{|\mathcal{M}|} \sum_{r \in \mathcal{M}} \text{DTW}(s, r)$. We perform the same calculation for each generated sequence to obtain $d_{\mathcal{G}}(s)$. This creates two collections of mean distances across all choices of $s$, which we convert into empirical distributions $D_{\mathcal{R}}$ and $D_{\mathcal{G}}$. We then apply Jensen-Shannon divergence to compare these distance distributions:

$$\text{DTW-JS}(D_{\mathcal{R}}, D_{\mathcal{G}}) = \frac{1}{2}[\text{KL}(D_{\mathcal{R}}||D_M) + \text{KL}(D_{\mathcal{G}}||D_M)] \tag{14}$$

where $D_M = \frac{1}{2}(D_{\mathcal{R}} + D_{\mathcal{G}})$ is the mixture of the two distance distributions. Small DTW-JS values indicate that the real and generated samples are "distributionally" similar in terms of their temporal patterns. More details can be found in Appendix C.2.

## 4.2 Short Sequence Time Series Unconditional Generation

We follow the evaluation setup of TimeGAN (Yoon et al., 2019) and Diffusion-TS (Yuan & Qiao, 2024) to assess generation quality against baseline models. All datasets are segmented into sequences of length 24 using a sliding window with stride 1. For evaluation, we generate samples matching the size of the original training data for each dataset to ensure fair evaluation. Training configurations and times, as well as model complexity, are discussed in Appendices C.6 and C.7.

As shown in Table 1, our method consistently outperforms all diffusion-based baseline methods across most datasets and metrics. While FourierDiffusion achieves competitive performance on certain datasets, Wavelet-Diff demonstrates superior and more consistent results across all evaluation scenarios. In our evaluations, we also tested different wavelet families and selected Symlets wavelets for Stocks, Coiflets wavelets for fMRI, and Daubechies wavelets for other datasets. In general, even when universally adopting Daubechies wavelets, WaveletDiff outperforms other diffusion paradigms. More details regarding the influence of the wavelet basis function on generative performance are available in Appendix C.3. To highlight our model's ability to

capture real data distributions, we present t-SNE embeddings and probability density plots all six datasets Appendix D.1.

When compared against MSDformer (Feng et al., 2025), a transformer-based model operating on VQ-VAE latent representations, results are more mixed: MSDformer performs better on ETTh1, ETTh2, and especially fMRI, while WaveletDiff performs better on Exchange Rate and especially EEG. To better understand the significant performance gaps of WaveletDiff and MSDformer on fMRI and EEG, we investigate the underlying dataset characteristics in the wavelet domain that distinguish their generative performance.

Table 1: Time series generation performance comparison on short sequences (length 24) with training time (on one NVIDIA A100 GPU) and model parameter comparison between WaveletDiff and MSDformer.

| Metric | Methods | ETTh1 | ETTh2 | Stocks | Exchange Rate | fMRI | EEG |
|---|---|---|---|---|---|---|---|
| Discriminative Score (Lower the Better) | WaveletDiff | 0.005±.005 | 0.008±.007 | **0.005±.004** | **0.004±.001** | 0.087±.077 | **0.006±.008** |
| | MSDformer | **0.004±.003** | **0.006±.004** | 0.011±.009 | 0.008±.006 | **0.008±.004** | 0.053±.042 |
| | FourierDiffusion | 0.019±.007 | 0.016±.006 | 0.024±.003 | 0.015±.009 | 0.196±.013 | 0.016±.007 |
| | Diffusion-TS | 0.071±.002 | 0.038±.008 | 0.087±.008 | 0.032±.002 | 0.188±.018 | 0.304±.177 |
| | TimeGAN | 0.127±.047 | 0.106±.035 | 0.091±.047 | 0.257±.070 | 0.499±.001 | 0.161±.063 |
| | SigDiffusions | 0.353±.023 | 0.381±.048 | 0.371±.027 | 0.324±.055 | 0.482±.018 | 0.500±.000 |
| | KoVAE | 0.189±.020 | 0.049±.015 | 0.040±.023 | 0.122±.010 | 0.479±.025 | 0.244±.090 |
| | TimesFM | 0.066±.017 | 0.024±.012 | 0.059±.015 | 0.008±.006 | 0.385±.188 | 0.009±.004 |
| | Chronos | 0.110±.017 | 0.046±.043 | 0.073±.034 | 0.010±.005 | 0.458±.079 | 0.049±.009 |
| Predictive Score (Lower the Better) | WaveletDiff | **0.119±.002** | **0.106±.004** | **0.037±.000** | 0.037±.002 | 0.100±.000 | **0.000±.000** |
| | MSDformer | 0.121±.000 | **0.106±.005** | **0.037±.000** | 0.036±.003 | **0.093±.002** | **0.000±.000** |
| | FourierDiffusion | 0.120±.005 | 0.111±.003 | **0.037±.000** | 0.040±.001 | 0.100±.000 | **0.000±.000** |
| | Diffusion-TS | 0.120±.004 | 0.107±.003 | **0.037±.000** | 0.037±.002 | 0.100±.000 | 0.001±.000 |
| | TimeGAN | 0.152±.015 | 0.128±.005 | 0.038±.000 | 0.064±.005 | 0.124±.002 | **0.000±.000** |
| | SigDiffusions | 0.131±.002 | 0.125±.003 | 0.040±.001 | 0.089±.006 | 0.105±.000 | **0.000±.000** |
| | KoVAE | 0.126±.001 | 0.112±.003 | **0.037±.000** | 0.040±.004 | 0.225±.019 | **0.000±.000** |
| | TimesFM | 0.121±.003 | 0.108±.003 | **0.037±.000** | 0.038±.001 | 0.103±.000 | **0.000±.000** |
| | Chronos | 0.121±.003 | 0.107±.004 | **0.037±.000** | 0.038±.001 | 0.103±.000 | **0.000±.000** |
| Context-FID Score (Lower the Better) | WaveletDiff | 0.020±.001 | 0.023±.002 | 0.018±.002 | **0.006±.000** | 0.104±.006 | **0.006±.000** |
| | MSDformer | **0.003±.000** | **0.004±.000** | **0.002±.000** | 0.007±.000 | **0.010±.000** | 0.017±.003 |
| | FourierDiffusion | 0.031±.002 | 0.024±.003 | 0.093±.010 | 0.054±.013 | 0.169±.005 | 0.012±.001 |
| | Diffusion-TS | 0.151±.007 | 0.054±.002 | 0.187±.016 | 0.056±.007 | 0.106±.003 | 0.017±.001 |
| | TimeGAN | 0.661±.041 | 0.157±.011 | 0.110±.012 | 0.660±.042 | 1.404±.114 | 0.018±.001 |
| | SigDiffusions | 2.413±.179 | 1.053±.099 | 3.494±.383 | 1.691±.157 | 6.576±.210 | 0.022±.001 |
| | KoVAE | 1.108±.079 | 0.211±.016 | 0.067±.005 | 0.177±.013 | 1.608±.050 | 0.018±.004 |
| | TimesFM | 0.389±.035 | 0.080±.013 | 0.149±.024 | 0.014±.000 | 12.262±.126 | 0.016±.002 |
| | Chronos | 1.080±.398 | 0.431±.129 | 0.325±.039 | 0.948±.297 | 8.910±.384 | 0.014±.001 |
| Correlational Score (Lower the Better) | WaveletDiff | 0.043±.008 | 0.083±.016 | 0.005±.003 | **0.060±.020** | 1.177±.031 | **1.811±.963** |
| | MSDformer | **0.038±.005** | **0.079±.016** | **0.004±.003** | 0.066±.012 | **0.764±.049** | 4.058±.258 |
| | FourierDiffusion | 0.046±.009 | 0.095±.016 | 0.013±.003 | 0.072±.019 | 1.184±.023 | 3.544±.626 |
| | Diffusion-TS | 0.051±.007 | 0.089±.022 | 0.009±.007 | 0.115±.016 | 1.382±.036 | 4.764±.107 |
| | TimeGAN | 0.202±.010 | 0.185±.015 | 0.053±.003 | 0.416±.018 | 29.562±.067 | 8.820±.121 |
| | SigDiffusions | 0.210±.010 | 0.430±.025 | 0.070±.005 | 0.943±.024 | 15.389±.064 | 4.389±.257 |
| | KoVAE | 0.163±.019 | 0.243±.049 | 0.061±.004 | 0.150±.026 | 7.394±.033 | 6.918±.157 |
| | TimesFM | 0.102±.018 | 0.094±.016 | 0.028±.007 | 0.075±.016 | 6.450±.127 | 3.633±.495 |
| | Chronos | 0.118±.028 | 0.131±.016 | 0.121±.019 | 0.185±.056 | 4.873±.043 | 3.039±.468 |
| DTW-JS distance (Lower the Better) | WaveletDiff | 0.101±.016 | **0.064±.014** | 0.106±.027 | **0.121±.029** | 0.191±.011 | **0.055±.011** |
| | MSDformer | **0.094±.023** | 0.077±.010 | **0.105±.008** | 0.127±.021 | **0.115±.024** | 0.605±.062 |
| | FourierDiffusion | 0.105±.022 | 0.073±.014 | 0.138±.024 | 0.130±.021 | 0.286±.042 | 0.067±.017 |
| | Diffusion-TS | 0.111±.020 | 0.087±.012 | 0.153±.019 | 0.139±.031 | 0.237±.042 | 0.220±.013 |
| | TimeGAN | 0.155±.030 | 0.097±.042 | 0.142±.028 | 0.231±.025 | 0.215±.037 | 0.632±.049 |
| | SigDiffusions | 0.259±.024 | 0.273±.034 | 0.377±.068 | 0.376±.036 | 0.693±.000 | 0.293±.125 |
| | KoVAE | 0.104±.018 | 0.075±.016 | 0.122±.025 | 0.155±.016 | 0.693±.000 | 0.595±.063 |
| | TimesFM | 0.148±.025 | 0.082±.013 | 0.118±.006 | 0.129±.030 | 0.693±.000 | 0.144±.021 |
| | Chronos | 0.127±.035 | 0.077±.018 | 0.115±.017 | 0.123±.026 | 0.693±.000 | 0.086±.018 |
| Training Time (h:m:s) | WaveletDiff | **9:11:06** | **9:00:45** | 4:19:59 | **5:44:06** | **7:52:16** | **8:22:59** |
| | MSDformer | 9:36:02 | 9:27:33 | 5:23:36 | 11:47:29 | 25:53:28 | 9:48:10 |
| Model Parameter (M) | WaveletDiff | **63.1** | **63.1** | 63.1 | **63.1** | **63.2** | **63.1** |
| | MSDformer | 72 | 72 | **55.9** | 72 | 83.9 | 72 |

**For Which Time Series Does Wavelet-Domain Diffusion Work the Best?**  Note that WaveletDiff outperforms pure time and frequency domain diffusion models, and consequently, the issues to be discussed in what follows similarly pertain to other diffusion models that operate in transform domains. Among the six datasets, fMRI and EEG represent two extremes in terms of the kurtosis of their wavelet coefficients across decomposition levels, shown in Figure 3. EEG exhibits highly non-Gaussian coefficient distributions, whereas the wavelet coefficients of fMRI are very close to Gaussian. This observation is consistent with prior

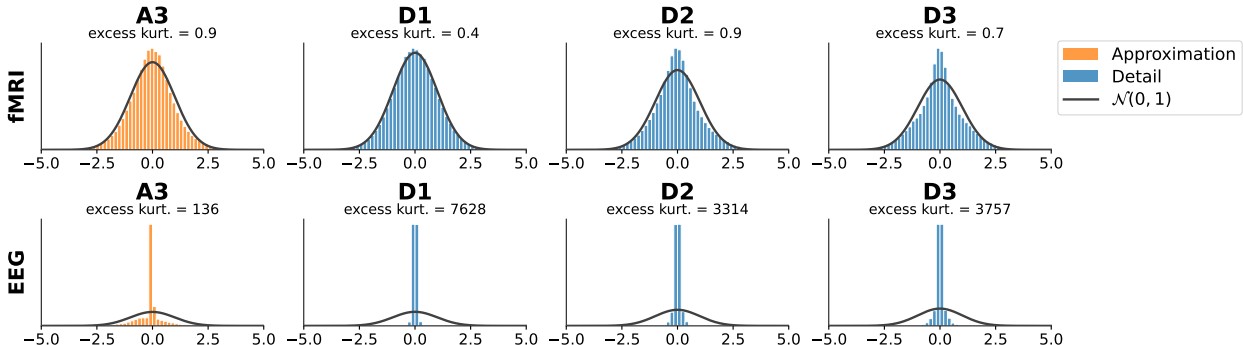

Figure 3: Standardized wavelet coefficient density of fMRI and EEG datasets. fMRI demonstrates near-Gaussian distributions while EEG exhibits highly non-Gaussian distributions. Energy distributions for coefficient at different levels are depicted in Figure 6 in the Appendix.

findings reported in the literature (Bullmore et al., 2004; Maxim et al., 2005) that pointed out that fMRI signals approximately follow a $1/f$ process, resulting in near-Gaussian wavelet coefficients for each level. The latter also explains why WaveletDiff performs relatively poorly on fMRI compared to MSDformer: The near-Gaussian coefficient distributions contain less higher-order statistical structure for our level-specific transformer architecture to exploit. In other words, the near-Gaussian distributions of the coefficients at each level render Gaussian diffusion redundant. On the other hand, for EEG signals, the wavelet coefficients exhibit distributions that exhibit more structure and are individually easier to learn, which makes them highly amenable for wavelet-domain diffusion.

To verify these findings even further, we added results on EEG-related time series recordings that exhibit similar wavelet-domain structure: Intracranial EEG (iEEG) (Nejedly et al., 2020) and scalp-EEG sleep recordings from Sleep-EDF (Kemp et al., 2000; Goldberger et al., 2000). As may be seen from Table 8 in the Appendix, WaveletDiff outperforms MSDformer on these types of series as expected. A further ablation study in Table 6 based on replacing level-specific transformers with one single shared transformer further supports our conclusions. Interestingly, the generative performance on fMRI improves on multiple metrics with a shared transformer, whereas the performance of other datasets generally degrades. This supports our hypothesis that level-specific modeling is less beneficial for near-Gaussian wavelet coefficients but clearly more effective for complex coefficient distributions.

**A Closer Look at the Context-FID Metric.** Although WaveletDiff performs comparably to MSD-Former on almost all datasets tested except fMRI and EEG, it shows a pronounced gap with respect to Context-FID metric. This can be attributed to the fundamental architectural difference between the two generation paradigms. WaveletDiff is a diffusion model that directly learns the mapping from a Gaussian prior to the data distribution, whereas MSDFormer generates samples using a discrete-token transformer operating on the latent manifold learned by a VQ-VAE. Since Context-FID is computed as the FID between TS2Vec embeddings of real and generated samples, MSDFormer benefits from generation within a learned latent manifold, naturally leading to lower Context-FID scores. In contrast, diffusion models generate samples directly from Gaussian noise without such manifold constraints. Therefore, Context-FID may not be fully adequate for comparing these fundamentally different generation paradigms. This interpretation is further supported by the fact that, although WaveletDiff underperforms MSDFormer on Context-FID, it consistently outperforms *all other diffusion-based models on the same metric.* It also suggests a new direction for further improving WaveletDiff through specialized vector quantization methods and manifold constraints.

### 4.3 Long Sequence Time Series Unconditional Generation

We assess the performance of WaveletDiff for long time series generation by segmenting datasets into sequences of length 32, 64, and 128, again using a sliding window with stride 1. The same evaluation protocol

is applied, where we generate samples matching the size of the original training data for each dataset. As seen in Table 2, WaveletDiff offers consistent performance improvements across most settings. Here we used the spectral energy preservation term based on Parseval's theorem with a loss weight $\lambda_{\text{energy}} = 0.3$ on ETTh1 and Exchange Rate, but not on Stocks due to its high volatility.

Table 2: Time series generation performance comparison on long sequences.

| Dataset | Metric | Length | WaveletDiff | FourierDiffusion | Diffusion-TS | TimeGAN | SigDiffusions |
|---|---|---|---|---|---|---|---|
| ETTh1 | Discriminative | 32 | **0.016±.001** | 0.030±.004 | 0.078±.003 | 0.128±.036 | 0.346±.033 |
| | Score | 64 | **0.028±.009** | 0.048±.004 | 0.079±.010 | 0.116±.088 | 0.294±.156 |
| | (Lower the Better) | 128 | **0.034±.037** | 0.113±.006 | 0.159±.006 | 0.299±.148 | 0.462±.035 |
| | Predictive | 32 | **0.119±.001** | **0.119±.005** | **0.119±.003** | 0.126±.009 | 0.129±.000 |
| | Score | 64 | **0.114±.007** | **0.114±.004** | 0.120±.004 | 0.125±.004 | 0.129±.002 |
| | (Lower the Better) | 128 | 0.113±.005 | **0.112±.007** | 0.116±.005 | 0.177±.015 | 0.129±.002 |
| | Context-FID | 32 | **0.038±.005** | 0.048±.003 | 0.204±.011 | 0.599±.044 | 2.875±.027 |
| | Score | 64 | **0.088±.005** | 0.135±.010 | 0.265±.012 | 0.978±.114 | 6.622±.354 |
| | (Lower the Better) | 128 | **0.256±.014** | 0.356±.021 | 0.805±.094 | 11.813±.851 | 11.596±.800 |
| | Correlational | 32 | **0.050±.004** | 0.056±.019 | 0.064±.014 | 0.118±.013 | 0.180±.013 |
| | Score | 64 | 0.054±.009 | **0.052±.006** | 0.059±.010 | 0.307±.015 | 0.200±.023 |
| | (Lower the Better) | 128 | **0.059±.021** | 0.072±.010 | 0.083±.004 | 1.098±.005 | 0.235±.015 |
| | DTW-JS | 32 | **0.095±.022** | 0.099±.035 | 0.113±.022 | 0.226±.019 | 0.235±.017 |
| | Distance | 64 | **0.095±.028** | 0.105±.031 | 0.123±.034 | 0.208±.017 | 0.199±.031 |
| | (Lower the Better) | 128 | **0.105±.035** | 0.134±.022 | 0.122±.017 | 0.262±.051 | 0.122±.021 |
| Stocks | Discriminative | 32 | **0.006±.004** | 0.022±.012 | 0.099±.012 | 0.197±.025 | 0.357±.027 |
| | Score | 64 | **0.007±.003** | 0.032±.018 | 0.099±.008 | 0.152±.020 | 0.324±.044 |
| | (Lower the Better) | 128 | **0.015±.008** | 0.086±.036 | 0.141±.011 | 0.270±.124 | 0.339±.007 |
| | Predictive | 32 | **0.037±.000** | **0.037±.000** | 0.038±.000 | **0.037±.000** | 0.040±.001 |
| | Score | 64 | **0.036±.000** | **0.036±.000** | 0.037±.000 | 0.038±.000 | 0.039±.000 |
| | (Lower the Better) | 128 | **0.036±.000** | 0.038±.000 | 0.037±.000 | 0.070±.007 | 0.040±.000 |
| | Context-FID | 32 | **0.026±.006** | 0.087±.007 | 0.256±.029 | 0.449±.042 | 3.403±.373 |
| | Score | 64 | **0.047±.005** | 0.151±.026 | 0.369±.065 | 0.336±.046 | 4.229±.495 |
| | (Lower the Better) | 128 | **0.080±.012** | 0.379±.025 | 0.417±.077 | 3.231±.325 | 5.472±.004 |
| | Correlational | 32 | **0.002±.002** | 0.011±.001 | 0.017±.007 | 0.094±.006 | 0.075±.004 |
| | Score | 64 | **0.003±.001** | 0.013±.005 | 0.020±.002 | 0.098±.003 | 0.052±.004 |
| | (Lower the Better) | 128 | **0.004±.002** | 0.162±.011 | 0.021±.006 | 0.621±.006 | 0.091±.004 |
| | DTW-JS | 32 | **0.112±.025** | 0.118±.021 | 0.137±.026 | 0.182±.026 | 0.301±.060 |
| | Distance | 64 | 0.136±.021 | 0.139±.008 | **0.136±.018** | 0.155±.031 | 0.261±.013 |
| | (Lower the Better) | 128 | 0.112±.013 | 0.127±.020 | **0.116±.004** | 0.420±.015 | 0.281±.058 |
| Exchange Rate | Discriminative | 32 | **0.011±.005** | 0.018±.013 | 0.031±.006 | 0.254±.064 | 0.314±.024 |
| | Score | 64 | **0.020±.005** | 0.038±.015 | 0.028±.005 | 0.277±.046 | 0.300±.007 |
| | (Lower the Better) | 128 | **0.026±.008** | 0.092±.032 | 0.046±.007 | 0.106±.064 | 0.276±.015 |
| | Predictive | 32 | **0.035±.002** | 0.040±.002 | 0.036±.002 | 0.069±.006 | 0.085±.007 |
| | Score | 64 | **0.035±.001** | 0.041±.001 | **0.035±.002** | 0.056±.005 | 0.078±.008 |
| | (Lower the Better) | 128 | **0.034±.003** | 0.044±.002 | **0.034±.002** | 0.048±.003 | 0.074±.005 |
| | Context-FID | 32 | **0.013±.001** | 4.057±.648 | 0.037±.003 | 1.038±.144 | 1.853±.164 |
| | Score | 64 | **0.022±.003** | 0.129±.065 | 0.056±.005 | 1.136±.114 | 1.834±.235 |
| | (Lower the Better) | 128 | **0.052±.003** | 0.264±.008 | 0.063±.004 | 0.849±.087 | 2.079±.168 |
| | Correlational | 32 | **0.064±.010** | 0.109±.037 | 0.091±.044 | 0.456±.016 | 1.065±.033 |
| | Score | 64 | **0.066±.024** | 0.096±.026 | 0.097±.012 | 0.421±.038 | 1.042±.027 |
| | (Lower the Better) | 128 | **0.065±.026** | 0.173±.011 | 0.101±.019 | 0.237±.035 | 1.001±.046 |
| | DTW-JS | 32 | **0.108±.037** | 0.116±.028 | 0.129±.023 | 0.182±.023 | 0.375±.045 |
| | Distance | 64 | **0.132±.015** | 0.142±.028 | 0.136±.010 | 0.195±.036 | 0.305±.024 |
| | (Lower the Better) | 128 | **0.124±.018** | 0.146±.029 | 0.145±.028 | 0.224±.020 | 0.306±.022 |

## 4.4 Reproducibility and Robustness to Irregular Sampling

Diffusion model analysis has been extended in many directions, including conditional generation, ambient diffusion (Daras et al., 2023), reproducibility and others. Many of these analyses have also been adapted to time series generation. In this work, we present the first study of reproducibility for time series diffusion models. Following recent diffusion model reproducibility studies (Zhang et al., 2024b; Li et al., 2024; Kadkhodaie et al., 2024), we investigate whether this phenomenon extends to the time series domain. Specifically, we train pairs of models with architectural variations and generate samples from identical Gaussian noise using deterministic DDIM sampling. Our results show that time series diffusion models exhibit strong reproducibility across all representation domains (see Appendix E). We further extend WaveletDiff to irregular sampled time series by following the preprocessing protocol of KoVAE and GT-GAN, where missing observations are first imputed using cubic spline interpolation before training. Experimental results demonstrate that WaveletDiff maintains consistently strong performance across different missing rates, indicating that the proposed wavelet-domain diffusion framework is robust to irregular sampling (see Appendix C.5).

### 4.5 Ablation Study

**Wavelet selection.** We analyzed the performance of the methods with respect to different wavelet design choices, including the wavelet basis, order, and level. We examined five different wavelet families and three different settings for the wavelet order and decomposition level. The ablation studies pertaining to the order and level are performed exclusively on Daubechies wavelets for all datasets, in order to ensure consistent comparisons. As shown in Table 4 in the Appendix, across the five wavelet families, performance varies with the type of datasets but consistently outperforms all baselines. We believe this to be the case because different wavelet families possess distinct properties, such as orthogonality, compact support, and smoothness, which directly shape coefficient sparsity and localization in the wavelet domain, making them better suited to different dataset characteristics. This ability to tailor the wavelet basis to the characteristics of a dataset represents a key advantage of WaveletDiff over frequency domain approaches which rely on fixed Fourier basis functions regardless of the underlying signal properties. More details can be found in Appendix C.3. In addition, as shown in Table 5 in the Appendix, different choices of wavelet order and decomposition level lead to minor performance variations, while all settings consistently outperform baselines.

**WaveletDiff architectural components.** To validate the effectiveness of different architectural components of WaveletDiff, we conduct an ablation study comparing our full model against six variants: (1) *Predicting coefficients rather than noise:* Instead of predicting noise as in standard DDPM, this variant directly predicts the wavelet coefficients themselves, following the approach in Diffusion-TS (Yuan & Qiao, 2024) which suggests this method outperforms noise prediction. (2) *Removing cross-attention:* This variant disables information exchange between different wavelet decomposition levels, but maintains level-specific transformer models. (3) *Using Cosine instead of Exponential schedules:* This variant replaces exponential with cosine noise scheduling during diffusion. (4) *Using DDIM sampling:* This variant uses deterministic DDIM rather than DDPM sampling during inference. (5) *Shared transformer:* Replacing the level-transformers to a single shared transformer. (6) *w/o larger approx-level capacity:* Reducing the number of layers and dimensions of the approximate level transformer to be the same as detail level transformer. The results in Table 6 in the Appendix reveal that *cross-level attention* is the universally most critical architectural component, with its removal causing discriminative scores and Context-FID scores to degrade on average by approximately $4\times$ and $3.5\times$, respectively. While cosine noise scheduling and DDIM sampling show competitive performance on certain datasets, they exhibit instability in neuroscience domain datasets fMRI and EEG. Additionally, the individual level-transformers structure and larger approximate level transformer also demonstrates critical improvement in most settings.

**Energy loss weight** As pointed out in Section 3.2, the base reconstruction loss is typically sufficient for short sequence generation since the spectral energy drift is minimal over limited temporal horizons. We only adopted energy-preserving regularization for long sequence generation on ETTh1 and Exchange Rate datasets. We found that setting the weight to 0.3 consistently offered good performance across different settings. As shown in Table 7 in the Appendix, the energy regularization term improves performance across most metrics, except for Exchange Rate with generation sequence length equal to 32, where the short-horizon, low-variance dynamics lead to negligible spectral energy drift, limiting the benefits of regularization.

### 4.6 Computational Analysis

WaveletDiff comprises approximately 63M trainable parameters and the training times across different datasets and sequence lengths are presented in Table 10 in the Appendix. Our model has larger computational times compared to diffusion-based baselines; however, for many time series generation applications, one most often prioritizes overall output quality over generation time. Also, note that generative models like ours are typically used in offline settings (e.g., augmentation, simulation), in which case latency is less critical than for forecasting. Also, time series are "less complex" than images or videos so that training completes within hours, while sampling thousands of sequences takes seconds. Given that generation quality is the primary bottleneck, we believe that the increased cost is justified by the substantial gains in fidelity (for details, see Appendix C.7). Furthermore, as shown in Table 1, WaveletDiff actually has a slightly smaller model size and shorter training time compared to the Transformer-based baseline MSDformer.

**Broader Impact Statement**

While WaveletDiff demonstrates strong empirical performance across diverse datasets and metrics, synthetic time series generation remains an imperfect process, and generated samples may not fully reproduce all characteristics of real-world data. Users should be cautious when deploying WaveletDiff-generated data on high-stakes downstream tasks such as clinical decision-making, financial risk assessment, or safety-critical engineering applications, where even small distributional mismatches between synthetic and real data can lead to unreliable and potentially dangerous outcomes. We encourage practitioners to evaluate generated data using domain-specific quality criteria in addition to the metrics considered in this work, and to view synthetic data as a complement to, rather than a substitute for, real-world data collection whenever feasible.

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

# A   Mother Wavelet Families

Different wavelet families provide distinct characteristics of multi-scale decompositions through their specific filter coefficients $\{h_k\}$ and $\{g_k\}$ in the two-scale relations (Equation 1). In our experiments, we used five representative wavelet families, each satisfying different relationships between their order $p$ and filter length $F$. The vanishing moment properties and definitions presented here follow the PyWavelets framework implementation (Lee et al., 2019).

## A.1   Daubechies Wavelets (db)

Daubechies wavelets of order $p$ (e.g., db-$p$) provide orthogonality, compact support, and exactly $p$ vanishing moments for the wavelet function $\psi(t)$, with filter length $F = 2p$. The scaling function $\phi(t)$ has zero vanishing moments for orthogonal wavelets. The filter coefficients $\{h_k\}_{k=0}^{F-1}$ are derived from polynomial factorization to maximize regularity and maintain compact support.

For db2 (Daubechies-2 with $p = 2$, $F = 4$), the low-pass filter coefficients equal:

$$h_0 = \frac{1+\sqrt{3}}{4\sqrt{2}}, \quad h_1 = \frac{3+\sqrt{3}}{4\sqrt{2}}, \tag{15}$$

$$h_2 = \frac{3-\sqrt{3}}{4\sqrt{2}}, \quad h_3 = \frac{1-\sqrt{3}}{4\sqrt{2}}. \tag{16}$$

The high-pass coefficients satisfy $g_k = (-1)^k h_{F-1-k}$. The orthogonality and vanishing moment conditions ensure that

$$\sum_{k=0}^{F-1} h_k = \sqrt{2}, \tag{17}$$

$$\sum_{k=0}^{F-1} h_k h_{k+2m} = \delta_{m,0}, \tag{18}$$

$$\sum_{k=0}^{F-1} k^j h_k = 0 \quad \text{for } j = 1, \ldots, p-1, \tag{19}$$

where $\delta_{m,0}$ is the Kronecker delta function, defined as:

$$\delta_{m,0} = \begin{cases} 1 & \text{if } m = 0 \\ 0 & \text{if } m \neq 0. \end{cases} \tag{20}$$

### A.2  Symlets (sym)

Symlets are modified Daubechies wavelets designed to improve symmetry while maintaining orthogonality and compact support. With a filter length $F = 2p$, Symlets have the same vanishing moment properties as Daubechies wavelets, i.e., $p$ vanishing moments for the wavelet function $\psi(t)$ and zero vanishing moments for the scaling function $\phi(t)$. They minimize an asymmetry measure $A$ that quantifies the deviation from perfect symmetry, namely

$$A = \sum_{k=0}^{F-1} k \cdot |h_k|^2 - \frac{F-1}{2} \sum_{k=0}^{F-1} |h_k|^2, \tag{21}$$

where the first term represents the weighted center of mass of the filter coefficients, while the second term represents the theoretical center for a perfectly symmetric filter. A lower value of $A$ indicates better symmetry.

For sym2 ($p = 2$, $F = 4$), the coefficients are optimized versions of db2 coefficients, satisfying the same orthogonality conditions but with improved phase linearity and near-symmetric properties for better temporal localization.

### A.3  Coiflets (coif)

Coiflets of order $p$ are designed with balanced vanishing moments for both the scaling $\phi(t)$ and wavelet function $\psi(t)$, with filter length $F = 6p$. The wavelet function $\psi(t)$ has $2p$ vanishing moments while the scaling function $\phi(t)$ has $2p-1$ vanishing moments, providing more balance for the moments when compared to Daubechies wavelets (in which case the scaling function has zero vanishing moments). The filter coefficients satisfy extended moment conditions of the form

$$\sum_{k=0}^{F-1} h_k = \sqrt{2}, \tag{22}$$

$$\sum_{k=0}^{F-1} k^j h_k = 0 \quad \text{for } j = 1, \ldots, 2p-1, \tag{23}$$

$$\sum_{k=0}^{F-1} g_k = 0, \tag{24}$$

$$\sum_{k=0}^{F-1} k^j g_k = 0 \quad \text{for } j = 1, \ldots, 2p. \tag{25}$$

For coif1 ($p = 1$, $F = 6$), the wavelet function has two vanishing moments and the scaling function has one vanishing moment. The six filter coefficients provide enhanced moment balancing between analysis and synthesis operations, with the scaling function having non-zero vanishing moments unlike orthogonal Daubechies wavelets.

## A.4 Biorthogonal Wavelets (bior)

Biorthogonal wavelets use different filters for decomposition and reconstruction, denoted as bior$p_r.p_d$ where $p_r$ and $p_d$ are the orders that determine the vanishing moment properties. For a general bior$p_r.p_d$ wavelet, we have the following:

- *Wavelet function $\psi(t)$*: $p_r$ vanishing moments;

- *Scaling function $\phi(t)$*: $p_d$ vanishing moments;

- *Filter lengths*: These depend on the specific bior$p_r.p_d$ configuration and are not given by simple formulas like those of other wavelet families.

For decomposition one uses a low-pass filter $\{h_k\}$ and a high-pass filter $\{g_k\}$, while for reconstruction one uses a low-pass filter $\{\tilde{h}_k\}$ (denoted with tilde) and a high-pass filter $\{\tilde{g}_k\}$. The tilde notation ˜ indicates the dual (reconstruction) filters that are different from the primal (decomposition) filters.

For bior2.2 ($p_r = p_d = 2$), both decomposition and reconstruction wavelet functions have two vanishing moments, and both scaling functions have two vanishing moments, which result in perfect symmetry. The filters also satisfy the perfect reconstruction condition,

$$\sum_k h_k \tilde{h}_{k+2m} + g_k \tilde{g}_{k+2m} = \delta_{m,0} \tag{26}$$

In the z-domain, where $H(z)$, $G(z)$, $\tilde{H}(z)$, and $\tilde{G}(z)$ are the z-transforms of the respective filter sequences, the perfect reconstruction condition is succinctly summarized as

$$H(z)\tilde{H}(z^{-1}) + H(-z)\tilde{H}(-z^{-1}) = 2. \tag{27}$$

## A.5 Reverse Biorthogonal Wavelets (rbio)

Reverse biorthogonal wavelets (rbio$p_r.p_d$) interchange the decomposition and reconstruction filter roles compared to standard biorthogonal wavelets, according to:

$$h_k^{\text{rbio}} = \tilde{h}_k^{\text{bior}} \tag{28}$$
$$g_k^{\text{rbio}} = \tilde{g}_k^{\text{bior}} \tag{29}$$

For reverse biorthogonal wavelets, the vanishing moment assignment follows the same pattern as biorthogonal wavelets:

- *Wavelet function $\psi(t)$* has $p_r$ vanishing moments.

- *Scaling function $\phi(t)$* has $p_d$ vanishing moments.

For rbio2.2 ($p_r = p_d = 2$), both the wavelet function $\psi(t)$ and scaling function $\phi(t)$ have two vanishing moments each, maintaining symmetric properties.

The choice of wavelet family affects the sparsity and localization properties of the decomposition, with symmetric wavelets (bior, rbio) providing better phase preservation, while orthogonal wavelets (db, sym) ensure energy conservation through orthogonality.

## B  Energy Regularization for Biorthogonal Wavelets

For biorthogonal and reverse biorthogonal wavelets, the discrete wavelet transform is not orthonormal, and the squared $\ell_2$ norm of the wavelet coefficients does not in general equal the signal energy. To arrive at the correct energy equations, let $\{\psi_i\}$ denote the primal (synthesis) wavelet basis and $\{\tilde{\psi}_i\}$ denote the corresponding dual (analysis) wavelet basis, satisfying the biorthogonality condition

$$\langle \psi_i, \tilde{\psi}_j \rangle = \delta_{ij}. \tag{30}$$

Given a signal $x$, define the analysis coefficients and the corresponding primal-basis projections as

$$c_i = \langle x, \tilde{\psi}_i \rangle, \qquad \tilde{c}_i = \langle x, \psi_i \rangle. \tag{31}$$

The signal energy then satisfies the *biorthogonal* Parseval identity

$$\|x\|_2^2 = \sum_i c_i \, \tilde{c}_i. \tag{32}$$

Applying this identity to wavelet coefficients at each decomposition level $l$, a theoretically consistent energy regularization term can be defined as

$$\mathcal{L}_{\text{bio-energy}} = \mathbb{E}\left[\sum_{l=1}^{L+1} \left| \sum_{n,j,k} \boldsymbol{C}_{n,j,k}^{(l)} \tilde{\boldsymbol{C}}_{n,j,k}^{(l)} - \sum_{n,j,k} \hat{\boldsymbol{C}}_{n,j,k}^{(l)} \widehat{\tilde{\boldsymbol{C}}}_{n,j,k}^{(l)} \right| \right], \tag{33}$$

where $\boldsymbol{C}^{(l)}$ denotes the standard analysis coefficients with respect to the dual (analysis) wavelet basis, and $\tilde{\boldsymbol{C}}^{(l)}$ denotes the corresponding projections onto the primal (synthesis) wavelet basis.

## C  Experimental Details

### C.1  Datasets

Table 3 lists detailed properties of the datasets used in our experiments, and their repository links.

Table 3: Summary of the dataset types and statistics.

| Dataset | # of Samples | Dim | Source |
|---|---|---|---|
| ETTh1 | 17420 | 7 | https://github.com/zhouhaoyi/ETDataset |
| ETTh2 | 17420 | 7 | https://github.com/zhouhaoyi/ETDataset |
| Stocks | 3685 | 6 | https://finance.yahoo.com/quote/GOOG |
| Exchange Rate | 7588 | 8 | https://github.com/laiguokun/multivariate-time-series-data |
| fMRI | 10000 | 50 | https://www.fmrib.ox.ac.uk/datasets/netsim |
| EEG | 14980 | 14 | https://archive.ics.uci.edu/dataset/264/eeg+eye+state |

### C.2  Metrics

**Discriminative Score.** The discriminative score captures how difficult it is for a classifier to distinguish between real and generated samples. The score is measured by $|\text{acc} - 0.5|$, where acc is the classification accuracy. A score close to 0 indicates that real and generated samples are indistinguishable to the classifier, while a score close to 0.5 indicates they are very different. We follow the setup of TimeGAN (Yoon et al., 2019) using a 2-layer GRU-based neural network as the classifier, trained with binary cross-entropy loss to distinguish between real (label=1) and synthetic (label=0) sequences.

**Predictive Score.** The predictive score captures how useful generated samples are for the forecasting task on real data. The score is measured by the mean absolute error (MAE) between predicted values and ground-truth values on test data. We follow TimeGAN (Yoon et al., 2019) using a 2-layer GRU-based sequence

predictor trained on synthetic data to predict the next time step features, evaluated on real sequences. Lower MAE values indicate better predictive utility of the generated samples.

**Context-FID.** (Paul et al., 2022) The Fréchet Inception Distance (FID) measures the distance between two multivariate Gaussian distributions, i.e.,

$$\text{FID}(X, Y) = ||\mu_X - \mu_Y||^2 + \text{Tr}(\Sigma_X + \Sigma_Y - 2(\Sigma_X \Sigma_Y)^{1/2}), \tag{34}$$

where $\mu_X, \mu_Y$ are the means and $\Sigma_X, \Sigma_Y$ are the covariance matrices of the two distributions. Context-FID adapts this to time series by replacing the Inception-v3 features with time series features. We follow Diffusion-TS (Yuan & Qiao, 2024) using TS2Vec (Yue et al., 2022) representations as the features. We extract embeddings from both real and generated sequences using a trained TS2Vec encoder, then compute FID in the embedding space. Lower Context-FID values indicate better distributional similarity.

**Correlational Score.** This metric assesses temporal dependencies by comparing cross-correlation matrices between real and generated data. For sequences with $D$ features, we compute the sample covariance matrix for each dataset, convert them to correlation matrices, and then measure the average absolute difference across all feature pairs according to

$$\text{Correlational Score} = \frac{1}{10} \sum_{i=1}^{D} \sum_{j=1}^{D} |\rho_{i,j}^{real} - \rho_{i,j}^{generated}|, \tag{35}$$

where $\rho_{i,j}^{real}$ and $\rho_{i,j}^{generated}$ are the correlation coefficients between features $i$ and $j$ for real and generated data, respectively. Note that we follow the Diffusion-TS (Yuan & Qiao, 2024) setup using the factor $\frac{1}{10}$, although $\frac{1}{D^2}$ could provide better normalization across different feature dimensions. The former choice of normalization ensures direct comparability with prior work.

**DTW-JS Distance.** We propose a Dynamic Time Warping Jensen-Shannon Distance (DTW-JS distance) metric, which combines DTW's temporal alignment capabilities with Jensen-Shannon divergence for distributional comparison. DTW computes the optimal alignment distance between two time series sequences $x$ and $y$ by minimizing

$$\text{DTW}(x, y) = \min_{\pi} \sum_{(i,j) \in \pi} |x_i - y_j| \tag{36}$$

where $x$ and $y$ are two time series sequences, $\pi$ represents a warping path consisting of index pairs $(i, j)$ that map elements from sequence $x$ to sequence $y$, and the path must satisfy DTW constraints: monotonicity (indices only increase), continuity (no skipping), and boundary conditions (path starts at $(1, 1)$ and ends at $(|x|, |y|)$). The warping allows sequences to be stretched or compressed along the time axis to find the best alignment, enabling DTW to handle sequences of different lengths and account for temporal shifts or speed variations between similar patterns.

For our metric, we create a reference set $\mathcal{M}$ by randomly sampling from both real samples $\mathcal{R}$ and generated samples $\mathcal{G}$ (i.e., by taking the union of samples of these two sets, and ensuring that both sets have the same number of elements). For each sample $s$ in the real set $\mathcal{R}$ and generated set $\mathcal{G}$, we compute its mean DTW distance to all samples in the reference set:

$$d(s) = \frac{1}{|\mathcal{M}|} \sum_{r \in \mathcal{M}} \text{DTW}(s, r) \tag{37}$$

This creates two collections of mean DTW distances, which we histogram across different samples $s$ to form distance distributions $D_{\mathcal{R}}$ and $D_{\mathcal{G}}$ for real and generated samples, respectively. We then apply Jensen-Shannon divergence to compute the distance between the two distance distributions, i.e.,

$$\text{DTW-JS}(D_{\mathcal{R}}, D_{\mathcal{G}}) = \frac{1}{2}[\text{KL}(D_{\mathcal{R}}||D_M) + \text{KL}(D_{\mathcal{G}}||D_M)] \tag{38}$$

where $D_M = \frac{1}{2}(D_{\mathcal{R}} + D_{\mathcal{G}})$ is the mixture of the two distance distributions. This approach measures distributional similarity between real and generated samples while accounting for temporal alignment flexibility, providing a robust evaluation metric that captures both temporal structure and statistical properties.

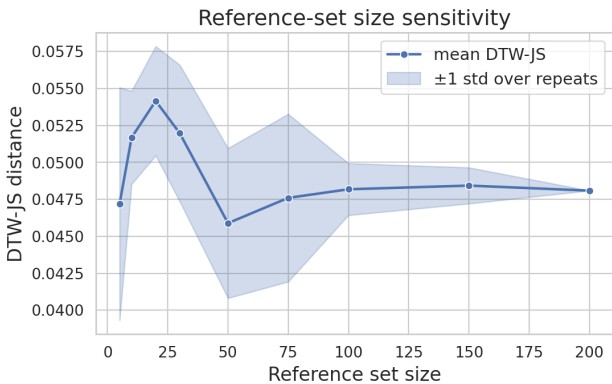

Figure 4: DTW-JS distance on EEG versus reference-set size $|\mathcal{M}|$. The metric is unstable for $|\mathcal{M}| < 50$ and stabilizes for $|\mathcal{M}| > 75$; we standardly use $|\mathcal{M}| = 100$.

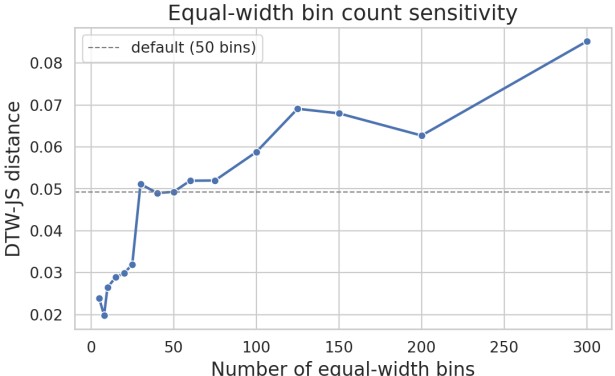

Figure 5: DTW-JS distance on EEG versus number of equal-width bins. Too few bins ($<30$) underestimate and too many ($>100$) inflate the distance; we standardly use 50 bins.

### C.2.1   Validation and Sensitivity Analysis of DTW-JS Distance

To validate the reliability of the proposed DTW-JS distance and characterize its behavior, we analyze its sensitivity to the two hyperparameters introduced in its definition, the reference-set size and the number of histogram bins; we also examine its computational cost. Unless otherwise stated, all analyses pertain to the EEG dataset as a representative example (on which WaveletDiff excels).

**Sensitivity to reference-set size.**   We randomly sample 100 sequences each from the generated and real datasets, and draw the reference set $M$ from their union. We vary the reference-set size from 5 to 200 samples. As shown in Figure 4, DTW-JS is unstable for small reference sets ($n < 50$) but converges and remains stable for larger sets ($n > 75$). In practice we use a reference set of 100 samples throughout our experiments.

**Sensitivity to histogram/binning size choices.**   By fixing the reference set at 100 samples, we analyze the sensitivity of DTW-JS to the number of bins used to approximate the distance distributions, with all bins of equal-width. As shown in Figure 5, too few bins ($< 30$) make the distributions overly coarse and systematically underestimate the DTW-JS distance, whereas too many bins ($> 100$) make them overly small and inflate the distance. We use 50 bins throughout, which yields a stable estimate, and apply the same setting when evaluating all baseline models to ensure a fair comparison.

Table 4: Mother wavelet selection.

| Metrics | Wavelet | ETTh1 | ETTh2 | Stocks | Exchange Rate | fMRI | EEG |
|---|---|---|---|---|---|---|---|
| Discriminative Score (Lower the Better) | db | **0.005±.005** | **0.008±.007** | 0.013±.007 | **0.004±.001** | 0.175±.071 | **0.006±.008** |
| | sym | 0.023±.005 | 0.023±.005 | **0.005±.004** | 0.011±.009 | 0.196±.066 | 0.007±.003 |
| | coif | 0.025±.010 | 0.031±.004 | 0.017±.009 | 0.073±.010 | **0.087±.077** | 0.014±.014 |
| | bior | 0.022±.008 | 0.033±.002 | 0.012±.004 | 0.051±.012 | 0.273±.007 | 0.008±.004 |
| | rbio | 0.057±.009 | 0.074±.008 | 0.010±.008 | 0.094±.010 | 0.129±.127 | 0.017±.009 |
| Predictive Score (Lower the Better) | db | 0.119±.002 | 0.106±.004 | **0.037±.000** | 0.037±.002 | **0.100±.000** | **0.000±.000** |
| | sym | 0.117±.004 | 0.107±.003 | **0.037±.000** | **0.035±.003** | **0.100±.000** | **0.000±.000** |
| | coif | **0.115±.005** | 0.106±.004 | **0.037±.000** | 0.036±.003 | **0.100±.000** | **0.000±.000** |
| | bior | 0.122±.002 | 0.109±.004 | **0.037±.000** | 0.037±.001 | **0.100±.000** | **0.000±.000** |
| | rbio | 0.121±.004 | **0.104±.001** | **0.037±.000** | 0.036±.001 | **0.100±.000** | **0.000±.000** |
| Context-FID Score (Lower the Better) | db | **0.020±.001** | **0.023±.002** | 0.024±.004 | **0.006±.000** | 0.104±.003 | **0.006±.000** |
| | sym | 0.052±.004 | 0.051±.006 | 0.018±.002 | 0.009±.001 | 0.122±.007 | 0.011±.001 |
| | coif | 0.079±.008 | 0.069±.009 | 0.018±.003 | 0.108±.008 | **0.104±.006** | **0.006±.000** |
| | bior | 0.049±.003 | 0.156±.016 | **0.016±.001** | 0.088±.013 | 0.119±.004 | **0.006±.001** |
| | rbio | 0.161±.005 | 0.225±.041 | **0.016±.002** | 0.175±.027 | 0.176±.008 | 0.007±.001 |
| Correlational Score (Lower the Better) | db | 0.043±.008 | 0.083±.016 | 0.006±.003 | **0.060±.020** | **1.073±.005** | **1.811±.963** |
| | sym | 0.055±.008 | 0.073±.025 | 0.005±.003 | 0.066±.012 | 1.172±.048 | 2.164±.533 |
| | coif | **0.036±.006** | **0.064±.011** | 0.007±.004 | 0.167±.032 | 1.177±.031 | 1.971±.969 |
| | bior | 0.048±.011 | 0.094±.009 | 0.005±.003 | 0.137±.020 | 1.147±.033 | 3.034±.759 |
| | rbio | 0.051±.008 | 0.099±.026 | **0.003±.004** | 0.161±.018 | 1.402±.034 | 1.959±.707 |
| DTW-JS Distance (Lower the Better) | db | **0.101±.016** | **0.064±.014** | 0.121±.013 | **0.121±.029** | 0.199±.043 | 0.055±.011 |
| | sym | 0.123±.009 | 0.067±.023 | **0.106±.027** | 0.132±.017 | 0.283±.035 | **0.049±.015** |
| | coif | 0.104±.015 | 0.086±.015 | 0.115±.016 | 0.157±.022 | **0.191±.011** | 0.062±.017 |
| | bior | 0.117±.013 | 0.095±.019 | 0.109±.009 | 0.129±.033 | 0.280±.011 | 0.050±.009 |
| | rbio | 0.110±.021 | 0.086±.033 | 0.119±.024 | 0.144±.037 | 0.464±.021 | 0.068±.022 |

**Computational cost** Under the default setting, with the real, generated, and reference sets each containing 100 samples, computing the DTW-JS distance once takes 5.77 seconds on average. Following the same protocol as for the other four metrics, we repeat the computation five times and report the mean and standard deviation, giving a total cost of roughly 30 seconds per evaluation.

## C.3 Wavelet Basis Function Analysis

The choice of mother wavelet significantly influences the multi-scale decomposition characteristics and subsequent series generation quality. Different wavelet families exhibit distinct properties in terms of orthogonality, compact support, and smoothness, which directly affect the sparsity and localization of coefficients in the wavelet domain. This choice becomes especially critical when dealing with diverse dataset characteristics, as different signal types require wavelets that can optimally capture their specific temporal-spectral patterns. As shown in Table 4, we systematically evaluated five representative wavelet families: Daubechies (db), Symlets (sym), Coiflets (coif), Biorthogonal (bior), and reverse Biorthogonal wavelets (rbio).

The results reveal dataset-specific preferred wavelets: Symlets work best for the Stocks dataset, likely due to their enhanced symmetry properties that better capture the near-symmetric fluctuations characteristic of financial time series. Coiflets demonstrate the best performance on the fMRI dataset, benefiting from balanced vanishing moments for both scaling and wavelet functions, which effectively capture the smooth yet complex spatiotemporal dynamics of brain activity. For the remaining datasets (ETTh1, ETTh2, Exchange Rate, EEG), Daubechies wavelets consistently provide the best overall performance. This ability to adapt the wavelet basis to match dataset characteristics represents an important advantage of WaveletDiff over frequency-domain approaches, which are constrained to use fixed Fourier basis functions regardless of the underlying signal properties, limiting their capacity to optimally represent diverse temporal patterns across different domains.

Table 5: Ablation study results for the wavelet order and decomposition level.

| Metrics | Order | Level | ETTh1 | ETTh2 | Stocks | Exchange Rate | fMRI | EEG |
|---|---|---|---|---|---|---|---|---|
| Discriminative | 2 | 3 | 0.005±.005 | **0.008±.007** | 0.013±.007 | **0.004±.001** | **0.175±.071** | **0.006±.008** |
| Score | 2 | 4 | 0.011±.006 | 0.011±.004 | **0.011±.006** | 0.006±.002 | 0.198±.090 | 0.007±.003 |
| (Lower the Better) | 1 | 3 | **0.004±.002** | 0.009±.001 | 0.016±.012 | **0.004±.003** | 0.298±.039 | 0.009±.003 |
| Predictive | 2 | 3 | 0.119±.002 | 0.106±.004 | **0.037±.000** | 0.037±.002 | **0.100±.000** | **0.000±.000** |
| Score | 2 | 4 | 0.120±.002 | 0.107±.003 | **0.037±.000** | **0.036±.003** | **0.100±.000** | **0.000±.000** |
| (Lower the Better) | 1 | 3 | **0.118±.004** | **0.104±.002** | **0.037±.000** | 0.037±.002 | 0.101±.000 | **0.000±.000** |
| Context-FID | 2 | 3 | 0.020±.001 | **0.023±.002** | 0.024±.004 | **0.006±.000** | **0.104±.003** | **0.006±.000** |
| Score | 2 | 4 | 0.034±.005 | 0.027±.001 | 0.025±.002 | 0.008±.000 | 0.113±.005 | 0.007±.001 |
| (Lower the Better) | 1 | 3 | **0.010±.001** | 0.041±.004 | **0.022±.002** | 0.008±.001 | 0.120±.004 | 0.013±.002 |
| Correlational | 2 | 3 | 0.043±.008 | 0.083±.016 | 0.006±.003 | 0.060±.020 | **1.073±.005** | **1.811±.963** |
| Score | 2 | 4 | 0.049±.009 | **0.078±.021** | **0.005±.004** | 0.048±.010 | 1.185±.023 | 2.455±.360 |
| (Lower the Better) | 1 | 3 | **0.033±.010** | 0.084±.008 | 0.009±.003 | 0.048±.010 | 1.287±.043 | 2.877±.698 |
| DTW-JS | 2 | 3 | 0.101±.016 | 0.064±.014 | 0.121±.013 | 0.121±.029 | 0.199±.043 | 0.055±.011 |
| Distance | 2 | 4 | **0.098±.026** | **0.060±.016** | **0.107±.016** | **0.114±.027** | 0.249±.017 | **0.048±.012** |
| (Lower the Better) | 1 | 3 | 0.101±.015 | 0.070±.018 | 0.117±.011 | 0.136±.039 | **0.165±.017** | 0.054±.008 |

Table 6: Ablation study results for key WaveletDiff architectural components.

| Metrics | Methods | ETTh1 | ETTh2 | Stocks | Exchange Rate | fMRI | EEG |
|---|---|---|---|---|---|---|---|
| | WaveletDiff | **0.005±.005** | **0.008±.007** | **0.005±.004** | **0.004±.001** | 0.087±.077 | **0.006±.008** |
| | coefficient prediction | 0.017±.013 | 0.040±.030 | 0.027±.014 | 0.059±.010 | 0.277±.012 | 0.020±.013 |
| Discriminative | w/o cross attention | 0.055±.054 | 0.028±.017 | 0.016±.012 | 0.052±.011 | 0.179±.016 | 0.006±.003 |
| Score | cosine noise scheduler | 0.024±.014 | 0.021±.019 | 0.094±.013 | 0.012±.005 | 0.112±.037 | 0.500±.000 |
| (Lower the Better) | DDIM sampling | 0.135±.094 | 0.017±.005 | 0.010±.003 | 0.006±.004 | 0.494±.006 | 0.024±.017 |
| | shared transformer | 0.500±.000 | 0.500±.000 | 0.007±.007 | 0.078±.011 | **0.064±.055** | 0.015±.010 |
| | w/o larger approx.-level capacity | 0.081±.005 | 0.071±.015 | 0.016±.007 | 0.153±.018 | 0.240±.014 | 0.017±.006 |
| | WaveletDiff | 0.119±.002 | **0.106±.004** | **0.037±.000** | 0.037±.002 | **0.100±.000** | **0.000±.000** |
| | coefficient prediction | 0.120±.003 | 0.111±.003 | **0.037±.000** | 0.038±.003 | **0.100±.000** | **0.000±.000** |
| Predictive | w/o cross attention | 0.119±.002 | **0.106±.002** | **0.037±.000** | 0.037±.001 | **0.100±.000** | **0.000±.000** |
| Score | cosine noise scheduler | 0.119±.003 | 0.107±.003 | **0.037±.000** | 0.037±.002 | 0.103±.000 | 0.172±.239 |
| (Lower the Better) | DDIM sampling | **0.118±.005** | 0.106±.004 | **0.037±.000** | **0.036±.002** | 0.101±.000 | **0.000±.000** |
| | shared transformer | 0.283±.043 | 0.256±.064 | **0.037±.000** | 0.037±.001 | **0.100±.000** | **0.000±.000** |
| | w/o larger approx.-level capacity | 0.127±.003 | 0.122±.003 | **0.037±.000** | 0.037±.002 | **0.100±.000** | **0.000±.000** |
| | WaveletDiff | **0.020±.001** | **0.023±.002** | 0.018±.002 | **0.006±.000** | 0.104±.006 | **0.006±.000** |
| | coefficient prediction | 0.027±.003 | 0.059±.004 | 0.053±.007 | 0.085±.008 | 0.131±.010 | 0.009±.001 |
| Context-FID | w/o cross attention | 0.118±.006 | 0.044±.004 | 0.039±.006 | 0.042±.004 | 0.170±.008 | 0.008±.001 |
| Score | cosine noise scheduler | 0.125±.005 | 0.050±.009 | 0.116±.021 | 0.011±.002 | 0.329±.022 | 44246±3637 |
| (Lower the Better) | DDIM sampling | 0.135±.006 | 0.049±.004 | **0.008±.002** | **0.006±.000** | 2.394±.043 | 0.009±.001 |
| | shared transformer | 2177736 | 1210986 | 0.015±.004 | 0.072±.002 | 0.130±.002 | 0.011±.001 |
| | w/o larger approx.-level capacity | 0.383±.039 | 0.352±.024 | 0.037±.005 | 0.247±.015 | 0.154±.012 | 0.007±.001 |
| | WaveletDiff | **0.043±.008** | 0.083±.016 | **0.005±.003** | **0.060±.020** | 1.177±.031 | 1.811±.963 |
| | coefficient prediction | 0.056±.010 | 0.097±.029 | 0.006±.003 | 0.182±.019 | 2.005±.051 | 2.650±.314 |
| Correlational | w/o cross attention | 0.046±.016 | 0.078±.016 | 0.006±.004 | 0.092±.035 | 1.755±.062 | 1.993±.738 |
| Score | cosine noise scheduler | 0.050±.014 | 0.105±.026 | 0.029±.010 | 0.070±.009 | 2.175±.071 | 1.487±.579 |
| (Lower the Better) | DDIM sampling | 0.057±.011 | 0.094±.018 | **0.005±.003** | 0.080±.031 | 2.113±.044 | **0.844±.150** |
| | shared transformer | 0.526±.008 | 0.627±.005 | 0.011±.005 | 0.168±.023 | **1.085±.025** | 3.762±.790 |
| | w/o larger approx.-level capacity | 0.063±.008 | **0.073±.012** | 0.010±.002 | 0.150±.019 | 1.852±.058 | 2.290±.929 |
| | WaveletDiff | **0.101±.016** | **0.064±.014** | **0.106±.027** | **0.121±.029** | 0.191±.011 | 0.055±.011 |
| | coefficient prediction | 0.115±.017 | 0.103±.030 | 0.133±.021 | 0.140±.024 | 0.204±.029 | 0.071±.019 |
| DTW-JS | w/o cross attention | 0.116±.029 | 0.085±.013 | 0.127±.020 | 0.133±.025 | 0.348±.009 | 0.064±.007 |
| Distance | cosine noise scheduler | 0.102±.018 | 0.075±.007 | 0.127±.021 | 0.134±.010 | 0.357±.033 | 0.693±.000 |
| (Lower the Better) | DDIM sampling | 0.112±.020 | 0.074±.018 | 0.113±.011 | 0.122±.022 | 0.693±.000 | **0.040±.014** |
| | shared transformer | 0.693±.000 | 0.693±.000 | 0.119±.021 | 0.160±.016 | **0.107±.017** | 0.074±.016 |
| | w/o larger approx.-level capacity | 0.138±.044 | 0.143±.020 | 0.121±.020 | 0.166±.013 | 0.213±.029 | 0.073±.026 |

## C.4 Wavelet Configuration Selection

We decsribe next how the three key wavelet hyperparameters — family, order, and decomposition level — are selected in WaveletDiff. While these can be tuned per dataset, our default heuristics yield strong performance without dataset-specific tuning.

Table 7: Ablation study regarding energy weights across different sequence lengths.

| Dataset | Length | Energy Weight | Discriminative Score | Predictive Score | Context-FID Score | Correlational Score | DTW-JS Distance |
|---|---|---|---|---|---|---|---|
| ETTh1 | 32 | 0 | 0.022±.015 | 0.117±.003 | 0.098±.009 | 0.054±.010 | 0.114±.015 |
| | | 0.3 | **0.016±.004** | **0.119±.001** | **0.038±.005** | **0.050±.004** | **0.095±.022** |
| | 64 | 0 | **0.027±.017** | **0.108±.004** | 0.168±.009 | 0.064±.022 | 0.111±.013 |
| | | 0.3 | 0.028±.009 | 0.114±.007 | **0.088±.005** | **0.054±.009** | **0.095±.028** |
| | 128 | 0 | 0.044±.026 | 0.121±.007 | 0.382±.022 | **0.056±.010** | 0.099±.007 |
| | | 0.3 | **0.034±.037** | **0.113±.005** | **0.256±.014** | 0.059±.021 | **0.105±.035** |
| Exchange Rate | 32 | 0 | **0.011±.005** | **0.035±.002** | **0.013±.001** | **0.064±.010** | **0.108±.037** |
| | | 0.3 | 0.032±.013 | 0.036±.001 | 0.057±.006 | 0.088±.025 | 0.126±.016 |
| | 64 | 0 | 0.128±.007 | 0.036±.002 | 0.432±.041 | 0.191±.052 | 0.209±.025 |
| | | 0.3 | **0.020±.005** | **0.035±.001** | **0.022±.003** | **0.066±.024** | **0.132±.015** |
| | 128 | 0 | 0.166±.017 | 0.036±.002 | 0.812±.146 | 0.217±.016 | 0.184±.008 |
| | | 0.3 | **0.026±.008** | **0.034±.003** | **0.052±.003** | **0.065±.026** | **0.124±.018** |

Table 8: WaveletDiff v.s. MSDformer on short sequences (length 24) generation on iEEG and Sleep-EDF.

| Metric | Methods | iEEG | Sleep-EDF |
|---|---|---|---|
| Discriminative Score | WaveletDiff | **0.009±.008** | **0.004±.004** |
| | MSDformer | 0.160±.041 | 0.006±.003 |
| Predictive Score | WaveletDiff | **0.024±.002** | **0.064±.000** |
| | MSDformer | 0.069±.001 | **0.064±.000** |
| Context-FID Score | WaveletDiff | **0.006±.000** | 0.043±.002 |
| | MSDformer | 0.826±.140 | **0.001±.000** |
| Correlational Score | WaveletDiff | **0.000±.000** | **0.000±.000** |
| | MSDformer | **0.000±.000** | **0.000±.000** |
| DTW-JS distance | WaveletDiff | **0.136±.016** | **0.091±.027** |
| | MSDformer | 0.230±.026 | 0.099±.019 |

**Wavelet Family.** The default wavelet family in WaveletDiff is the Daubechies wavelet family, which achieves consistently strong performance across all datasets (see Table 4). The ability to select different wavelet families for different signals is an additional feature of WaveletDiff that can lead to further improvements in generative performance on specific datasets. In our experiments, dataset-specific wavelet selection was performed by choosing the wavelet family with the best validation performance, which requires additional hyperparameter tuning and computational cost. However, the results in Table 4 demonstrate that such tuning is not really necessary in order to obtain strong performance in general, as the default Daubechies wavelet family already performs competitively across all benchmarks. It is important to notice that rigorous mathematical results are readily available to suggest which wavelet families are most amenable for which types of time series data decomposition/analysis (i.e., stationarity/nonstationarity, smoothness, sparseness etc) (Wojtaszczyk, 1997; Mallat, 2008). This is nevertheless not directly translatable into amenability for wavelet-domain generation, as seen on the example of fMRI data from the main text.

**Wavelet Order.** The wavelet order is selected based on the sequence length to avoid excessive boundary artifacts and ensure a meaningful multilevel decomposition. Specifically, for Daubechies wavelets, we use order $p = 2$ for sequence lengths $T \leq 32$, $p = 4$ for $33 \leq T \leq 64$, $p = 6$ for $65 \leq T \leq 128$, and $p = 8$ otherwise.

**Decomposition Level.** The decomposition level is based on the sequence length, wavelet family, and wavelet order. Following the PyWavelets framework, we compute the maximum decomposition level as $\lfloor \log_2(\frac{T}{F-1}) \rfloor$. We then clip the resulting value to the range $[3, 7]$ to avoid excessively shallow or deep decompositions, which would result in too few or too many wavelet-level transformer modules, respectively.

We further evaluate custom selections of wavelet order and decomposition levels (Table 5). The results show that moderate changes to these wavelet configurations lead to similar performance, indicating that WaveletDiff is relatively robust to the specific choice of wavelet parameters.

**Wavelet Implementation Details.** We use the PyWavelets implementation for all DWT and IDWT operations. Boundary effects are handled using symmetric extension. We follow the standard normalization conventions of the selected wavelet family as implemented in PyWavelets. Since we are using the symmetric extension, the energy regularizer is heuristic rather than exact.

## C.5 Irregular Sampling

We further evaluate WaveletDiff under the irregular sampling setting, where a subset of time steps is randomly removed from each training window before model training. Following the preprocessing protocol of KoVAE (Naiman et al., 2024c) and GT-GAN (Jeon et al., 2022), we use cubic spline interpolation to impute the missing values and then train WaveletDiff on the interpolated time series. As shown in Table 9, WaveletDiff maintains consistently strong performance across all missing rates.

Table 9: Irregular sampling with different missing rates $(30\%, 50\%, 70\%)$.

| Dataset | Missing Rate | Discriminative Score | Predictive Score | Context-FID Score | Correlational Score | DTW-JS Distance |
|---|---|---|---|---|---|---|
| ETTh1 | 0 (Regular) | 0.005±.005 | 0.119±.002 | 0.020±.001 | 0.043±.008 | 0.101±.016 |
| | 0.3 | 0.014±.002 | 0.120±.003 | 0.019±.001 | 0.064±.029 | 0.115±.030 |
| | 0.5 | 0.009±.006 | 0.121±.004 | 0.028±.004 | 0.045±.005 | 0.102±.026 |
| | 0.7 | 0.015±.009 | 0.121±.002 | 0.078±.013 | 0.056±.008 | 0.082±.011 |
| ETTh2 | 0 | 0.008±.007 | 0.106±.004 | 0.023±.002 | 0.083±.016 | 0.064±.014 |
| | 0.3 | 0.033±.002 | 0.106±.004 | 0.122±.011 | 0.109±.013 | 0.087±.019 |
| | 0.5 | 0.013±.007 | 0.105±.003 | 0.032±.003 | 0.072±.011 | 0.077±.019 |
| | 0.7 | 0.016±.007 | 0.108±.002 | 0.049±.002 | 0.073±.015 | 0.095±.013 |
| Stocks | 0 | 0.005±.004 | 0.037±.000 | 0.018±.002 | 0.005±.003 | 0.106±.027 |
| | 0.3 | 0.009±.007 | 0.037±.000 | 0.022±.003 | 0.007±.004 | 0.116±.022 |
| | 0.5 | 0.020±.008 | 0.037±.000 | 0.018±.003 | 0.005±.003 | 0.118±.012 |
| | 0.7 | 0.011±.005 | 0.037±.000 | 0.027±.003 | 0.006±.004 | 0.131±.016 |
| Exchange Rate | 0 | 0.004±.001 | 0.037±.002 | 0.006±.000 | 0.060±.020 | 0.121±.029 |
| | 0.3 | 0.011±.005 | 0.035±.001 | 0.010±.001 | 0.089±.039 | 0.119±.014 |
| | 0.5 | 0.019±.008 | 0.036±.002 | 0.030±.002 | 0.094±.036 | 0.129±.026 |
| | 0.7 | 0.010±.007 | 0.036±.001 | 0.010±.001 | 0.067±.019 | 0.136±.015 |
| fMRI | 0 | 0.087±.077 | 0.100±.000 | 0.104±.006 | 1.177±.031 | 0.191±.043 |
| | 0.3 | 0.304±.018 | 0.100±.000 | 0.371±.019 | 1.196±.038 | 0.605±.033 |
| | 0.5 | 0.344±.033 | 0.101±.000 | 1.109±.022 | 1.512±.067 | 0.564±.028 |
| | 0.7 | 0.380±.016 | 0.102±.000 | 1.407±.063 | 1.781±.071 | 0.318±.017 |
| EEG | 0 | 0.006±.008 | 0.000±.000 | 0.006±.000 | 1.811±.963 | 0.055±.011 |
| | 0.3 | 0.016±.009 | 0.000±.000 | 0.008±.001 | 2.595±.428 | 0.057±.008 |
| | 0.5 | 0.010±.007 | 0.000±.000 | 0.010±.001 | 3.683±1.025 | 0.054±.016 |
| | 0.7 | 0.009±.005 | 0.000±.000 | 0.007±.001 | 4.071±.797 | 0.043±.025 |

## C.6 Training Configuration Details

We provide training configuration information needed for reproducibility. The WaveletDiff model uses an embedding dimension of 256 with 8 attention heads across 8 transformer layers, a time embedding dimension of 128, and dropout rate of 0.1. The approximation level transformer uses twice as many embedding dimensions (512) and 10 layers to capture the critically important low-frequency information.

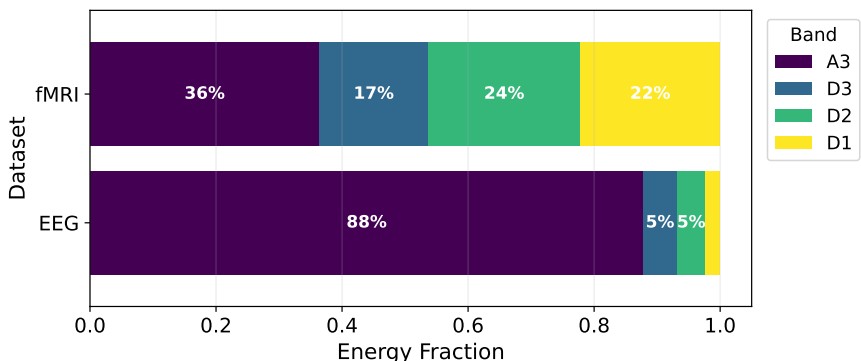

Figure 6: Energy distributions for wavelet coefficients at different levels of fMRI and EEG datasets. This has potential implications on how to select appropriate transformer structure for the levels.

The diffusion process employs 1000 timesteps with an exponential noise schedule. The exponential noise schedule is of the form

$$\beta_t = \beta_{start} + (\beta_{end} - \beta_{start}) \cdot \left(1 - e^{-\gamma \cdot t}\right) \tag{39}$$

where $\beta_{start} = 0.0001$, $\beta_{end} = 0.02$, $\gamma = 2.0$ is the exponential decay rate, $t \in [0, 1]$ is the normalized timestep, and $T = 1000$ is the total number of timesteps. The coefficient-weighted loss strategy assigns an approximation coefficient weight of 2.0 to emphasize low-frequency components. We train for 5000 epochs with batch size of 512.

For optimization, we use the AdamW optimizer with initial learning rate $2 \times 10^{-4}$ and weight decay $1 \times 10^{-5}$. We employ a one-cycle learning rate schedule (Smith & Topin, 2018) with cosine annealing strategy. The learning rate follows a two-phase schedule: Linear warm-up for the first 30% of training, then cosine annealing for the remaining 70%:

$$lr(e) = \begin{cases} lr_{base} + (lr_{max} - lr_{base}) \times \frac{e}{p \times E}, & \text{if } e \leq p \times E; \\ lr_{final} + (lr_{max} - lr_{final}) \times \left(\frac{1 + \cos\left(\pi \times \frac{e - p \times E}{(1-p) \times E}\right)}{2}\right), & \text{if } e > p \times E. \end{cases} \tag{40}$$

where $e$ is the current epoch, $lr_{base} = 4 \times 10^{-5}$, $lr_{max} = 1 \times 10^{-3}$, $lr_{final} = 4 \times 10^{-9}$, $p = 0.3$, and $E = 5000$ total epochs.

### C.7 Computational Setup and Training Time Analysis

WaveletDiff experiments were conducted on a single NVIDIA H100 GPU with 80GB memory using PyTorch 2.7.1 with CUDA 11.8, with training performed for 5000 epochs. Our WaveletDiff model contains approximately 63M trainable parameters. FourierDiffusion baseline results were obtained on a Tesla T4 GPU due to hardware constraints, following their original configuration and training settings. We note that the difference in GPU hardware was unavoidable due to High-Performance Computing (HPC) resource availability and the PyTorch Lightning deployment used for WaveletDiff, and was not selected to give any unfair advantage to our model. Tables 10 and 11 present the training times for WaveletDiff and FourierDiffusion across different datasets and sequence lengths, respectively. These measurements are provided as implementation details and should not be interpreted as a direct comparison of computational efficiency, since the experiments were conducted on different hardware platforms. Overall, the reported runtimes demonstrate that WaveletDiff can be trained within a practical time budget on a single modern GPU, indicating that compute time is not a bottleneck. In addition, we also provided a comparison of training time and model size between WaveletDiff and MSDformer on the same device (NVIDIA A100) in Table 1. The results show that WaveletDiff has a slightly smaller model size and shorter training time than the Transformer-based baseline. We also showed the inference time of WaveletDiff in Table 12, demonstrating that sampling can be done in seconds.

Table 10: Training times (hours:minutes:seconds) for WaveletDiff across different datasets and sequence lengths on single NVIDIA H100 GPU.

| sequence length | ETTh1 | ETTh2 | Stocks | Exchange Rate | fMRI | EEG |
|---|---|---|---|---|---|---|
| 24 | 3:45:54 | 3:36:34 | 1:07:05 | 1:36:59 | 2:40:13 | 3:22:06 |
| 32 | 3:45:33 | 3:38:23 | 1:02:49 | 1:46:32 | 2:42:41 | 3:24:07 |
| 64 | 3:24:46 | 4:19:04 | 1:12:34 | 2:03:44 | 3:15:34 | 4:13:58 |
| 128 | 5:06:03 | 6:27:03 | 1:33:53 | 2:59:33 | 4:32:52 | 6:01:14 |

Table 11: Training times (hours:minutes:seconds) for FourierDiffusion across different datasets and sequence lengths on single NVIDIA Tesla T4 GPU.

| sequence length | ETTh1 | ETTh2 | Stocks | Exchange Rate | fMRI | EEG |
|---|---|---|---|---|---|---|
| 24 | 1:09:51 | 1:19:01 | 1:57:23 | 2:21:39 | 2:21:38 | 2:39:03 |
| 32 | 1:38:56 | 1:39:13 | 2:10:41 | 2:30:54 | 2:27:28 | 3:47:09 |
| 64 | 4:49:37 | 4:49:39 | 3:14:13 | 2:31:45 | 3:58:26 | 5:21:55 |
| 128 | 10:14:05 | 10:14:04 | 6:35:18 | 3:46:43 | 6:29:26 | 7:16:49 |

## D  Visualization

### D.1  T-SNE and Data Distribution on Short Sequence Generation

We present the t-SNE and data distribution visualization of short sequence generation on ETTh1, ETTh2, Stocks, Exchange Rate, fMRI, and EEG dataset in Figure 7, 8, 9, 10, 11, and 12, respectively. The results demonstrate that WaveletDiff consistently outperforms diffusion baseline methods in capturing the underlying data distributions across all datasets.

## E  Diffusion Model Reproducibility Analysis

Inspired by recent work on reproducibility in diffusion models for images (Zhang et al., 2024b; Li et al., 2024; Kadkhodaie et al., 2024), we examine whether this phenomenon extends to time series generation. Specifically, we train pairs of models with slightly different architectures or configurations, then generate samples from the same fixed Gaussian noise input using deterministic DDIM sampling. For each pair of generated sequences $(x_1, x_2)$, we compute their similarity using dynamic time warping (DTW) distance. To quantify reproducibility, we follow (Zhang et al., 2024b) and define the RP score as

$$\text{RP score} := \mathbb{P}\big(\text{DTW}(x_1, x_2) < \overline{\text{DTW}}_{\text{rand}}\big), \tag{41}$$

where $\overline{\text{DTW}}_{\text{rand}}$ denotes the average DTW distance between randomly chosen sequence pairs generated by the two models. Thus, the RP score measures the probability that two models produce more similar samples from the same noise than would be expected by chance. An RP score greater than 0.5 indicates reproducibility.

Unlike image generation, which commonly uses U-Net architectures for comparison, time series generation employs diverse architectures. We examine the RP score across different model variations for both WaveletDiff and FourierDiffusion. Table 13 demonstrates that reproducibility exists for time series data regardless

Table 12: Inference time (seconds per 1,000 samples) for WaveletDiff across different datasets on single NVIDIA H100 GPU.

| | ETTh1 | ETTh2 | Stocks | Exchange Rate | fMRI | EEG |
|---|---|---|---|---|---|---|
| Time (sec) | 35 | 25 | 27 | 27 | 33 | 23 |

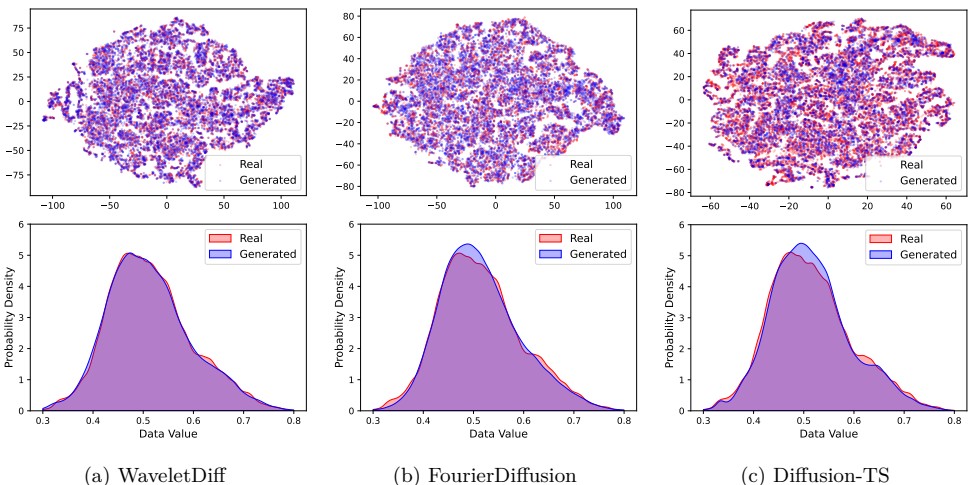

Figure 7: t-SNE visualization and probability distributions of training/synthetic data for ETTh1.

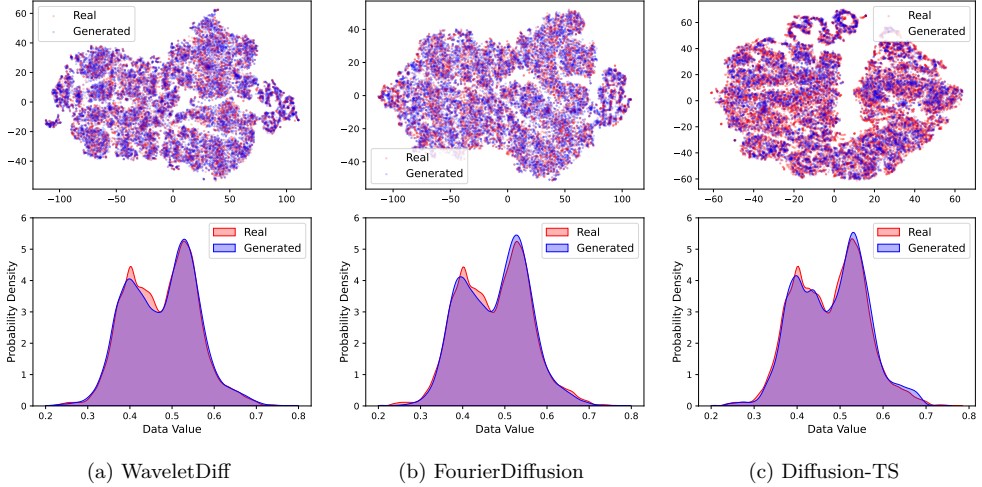

Figure 8: t-SNE visualization and probability distribution of data values on ETTh2 dataset.

of the representation domain (wavelet, time, or Fourier). To the best of our knowledge, we are the first to examine the reproducibility phenomenon specifically for time series generation.

Table 13: Reproducibility scores for different model variations demonstrate that time series diffusion models exhibit reproducibility across architectural changes and representation domains.

| Datasets | Model 1 | Model 2 | RP score |
|---|---|---|---|
| Stocks | WaveletDiff | WaveletDiff w/o cross-attention | 1.0 |
| | | WaveletDiff + cosine noise scheduler | 0.66 |
| | FourierDiffusion | FourierDiffusion on time domain | 0.665 |
| | | FourierDiffusion using LSTM score model | 0.805 |
| Exchange Rate | WaveletDiff | WaveletDiff w/o cross-attention | 0.999 |
| | | WaveletDiff + cosine noise scheduler | 0.619 |
| | FourierDiffusion | FourierDiffusion on time domain | 0.610 |
| | | FourierDiffusion using LSTM score model | 0.945 |

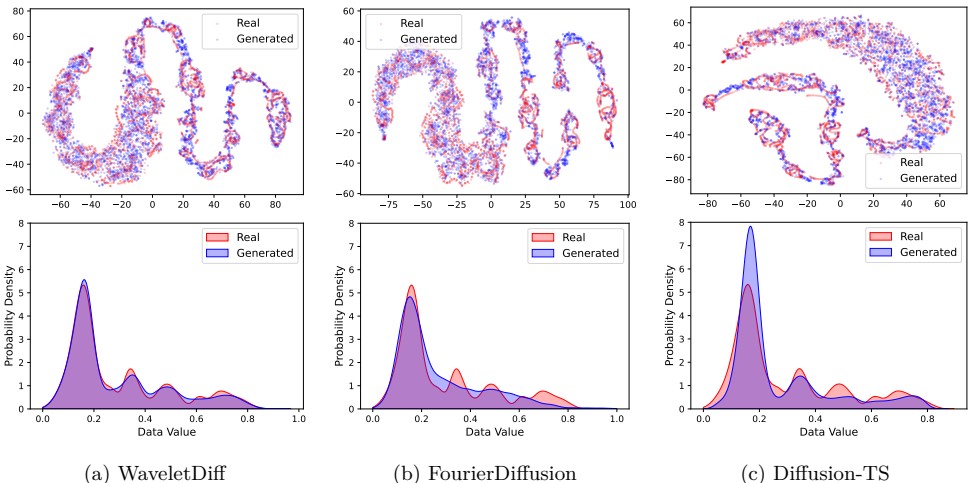

Figure 9: t-SNE visualization and probability distributions of training/synthetic data for Stocks.

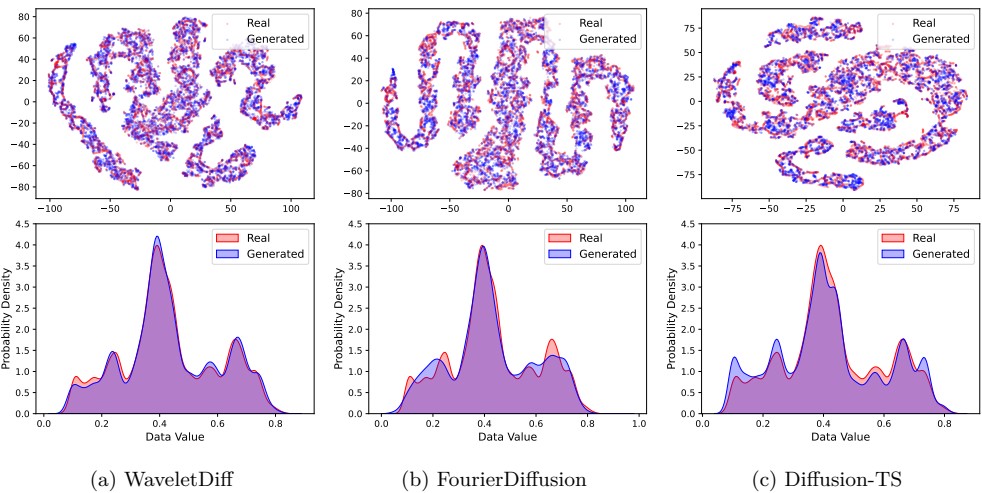

Figure 10: t-SNE visualization and probability distribution of data values on Exchange Rate dataset.

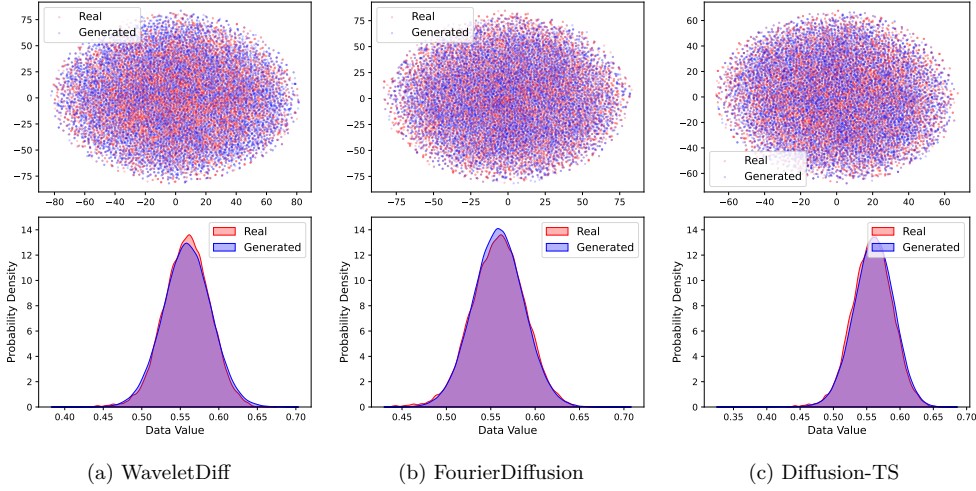

Figure 11: t-SNE visualization and probability distribution of data values on fMRI dataset.

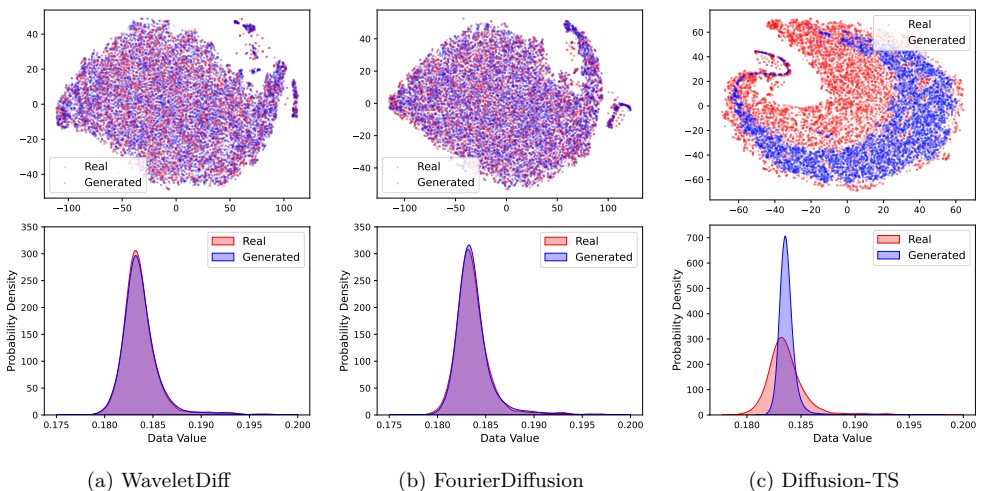

(a) WaveletDiff        (b) FourierDiffusion        (c) Diffusion-TS

Figure 12: t-SNE visualization and probability distribution of data values on EEG dataset.

## E.1 Impact of Architectural Variations

We evaluate how architectural modifications affect reproducibility by removing the cross-attention module from WaveletDiff. Figures 13 and 14 show that the model maintains strong reproducibility despite this significant architectural change, with generated sequences from identical noise exhibiting nearly identical patterns.

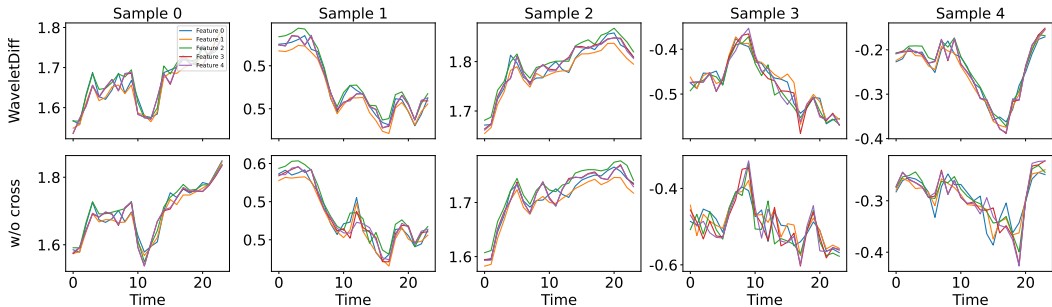

Figure 13: Reproducibility comparison on Stocks dataset using identical initial noise (volume feature excluded for clarity).

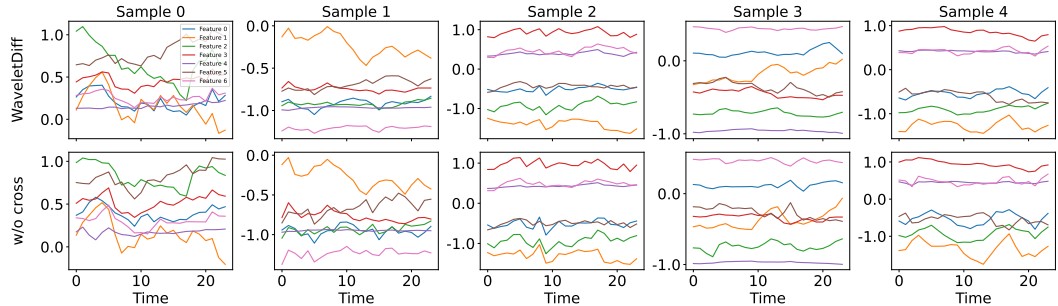

Figure 14: Reproducibility comparison on Exchange Rate dataset using identical initial noise.

### E.2 Impact of Mother Wavelet Selection

We further investigate how different mother wavelet choices affect reproducibility. As shown in Figures 15 and 16, Daubechies and Symlets wavelets demonstrate high reproducibility, while Coiflets, Biorthogonal, and Reverse Biorthogonal wavelets also exhibit good consistency. This indicates that wavelet choice significantly affects the learned distribution, with similar wavelet families (e.g., orthogonal wavelets like Daubechies and Symlets) producing more comparable results than dissimilar families.

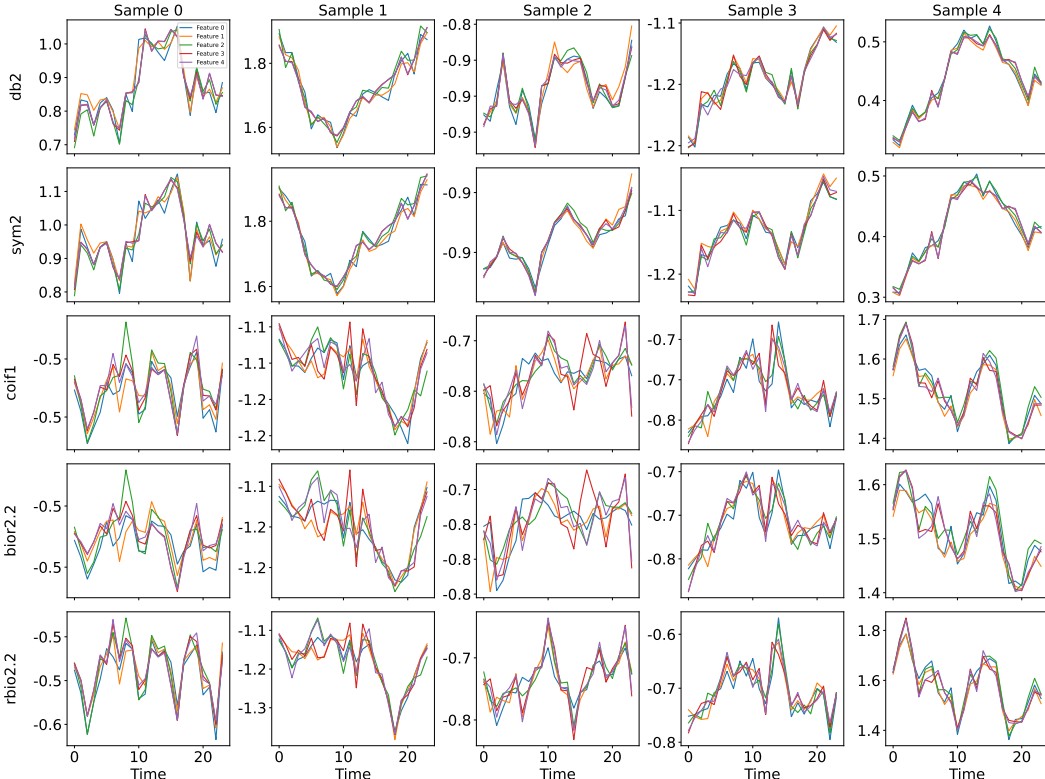

Figure 15: Wavelet family comparison on Stocks dataset demonstrating varying reproducibility across different mother wavelets.

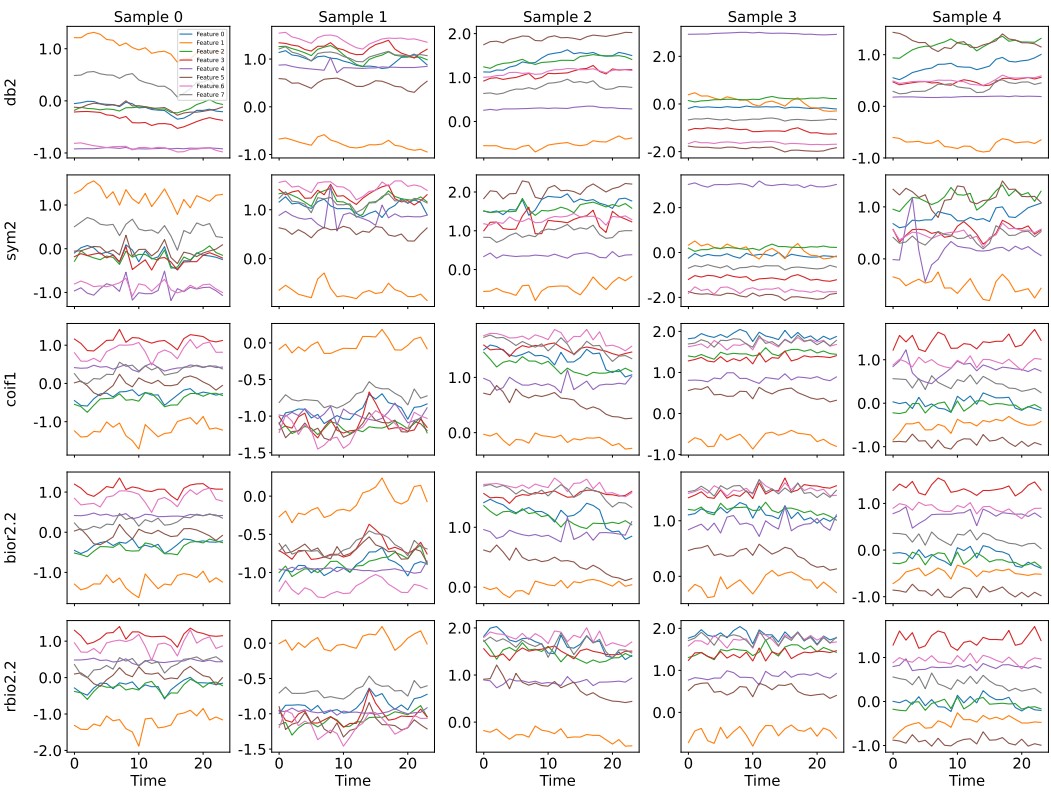

Figure 16: Wavelet family comparison on Exchange Rate dataset demonstrating varying reproducibility across mother wavelets.

