# OpenReview forum: "WaveletDiff: Multilevel Wavelet Diffusion For Time Series Generation"
_TMLR — Under review for TMLR_

### Review · Reviewer_YSHx · 2026-06-17

**Summary Of Contributions:**

WaveletDiff proposes a diffusion-based time series generation framework that operates directly on wavelet coefficients. By using level-specific transformers, cross-level attention, and energy-preserving constraints, it captures both local temporal details and global spectral structures. Experiments on real-world datasets show that WaveletDiff consistently outperforms strong time-domain and frequency-domain baselines, especially in discriminative and Context-FID scores.

**Audience:**

Yes

**Audience Explanation:**

The paper proposes a practical wavelet-domain diffusion framework and reports strong improvements over time-domain and Fourier-domain methods. The findings are relevant for people who need synthetic time series that preserve both local temporal details and global spectral structure.

**Broader Impact Concerns:**

No concerns.

**Claims And Evidence:**

Yes

**Claims Explanation:**

The main claims are supported by experiments on real-world datasets, multiple baselines and evaluation metrics, and ablation studies showing the importance of cross-level attention and energy regularization. However, I would still want clearer evidence on fairness issues such as dataset-specific wavelet selection and compute/parameter-matched comparisons.

**Requested Changes:**

* Clarify the wavelet selection protocol. The paper should explain how the wavelet family, order, and decomposition level were selected for each dataset, and on what basis this selection was made.
* Add compute or parameter-matched baseline comparisons. WaveletDiff has a relatively large model size and higher training cost; comparing it with stronger or similarly sized baselines can show that the gains are not mainly because of the larger capacity.
* Additional experiments on noisy, irregularly sampled, missing-value, or out-of-distribution time series would help demonstrate that WaveletDiff is robust beyond the clean benchmark setting.
* Explain and summarize computational trade-offs, such as training time, inference time, memory cost, and model size, in the main text, not only in the appendix. The computational cost is an important factor in showing the advantage or limitation of your work.

---

> ### Author Response · Authors · 2026-07-03
>
> We thank the reviewer for the insightful feedback. Below are our responses; these, in addition to more in-depth results and discussions, have been added to the revised text in blue.
>
> 1.
> **Wavelet Family**: The default wavelet family in WaveletDiff is the Daubechies wavelet family, which achieves consistently strong performance across all datasets (see Table 4 in the Appendix). The ability to select different wavelet families for different signals is an additional feature of WaveletDiff that can lead to further improvements in generative performance on specific datasets. In our experiments, dataset-specific wavelet selection was performed by choosing the wavelet family with the best validation performance, which requires additional hyperparameter tuning and computational cost. However, to point out once again, the results in Table 4 demonstrate that such tuning is not really necessary in order to obtain strong performance in general, as the default Daubechies wavelet family already performs competitively across all benchmarks. Furthermore, it is important to notice that rigorous mathematical results are readily available to suggest which wavelet families are most amenable for which types of time series data decomposition/analysis (i.e., stationarity/nonstationarity, smoothness, sparseness etc). We added a citation to these works [1, 2] but strongly emphasize the fact that these results pertain to signal analysis, and not signal generation. Analysis and generation are very different tasks and often, as we saw in our comparative studies, what makes a data class amenable for analysis hurts generation and vice versa (e.g., fMRI data).
>
> **Wavelet Order**: The wavelet order is selected based on the sequence length to avoid excessive boundary artifacts and ensure a meaningful multilevel decomposition. Specifically, for Daubechies wavelets, we use order 2 for sequence lengths ≤ 32, order 4 for lengths between 33 and 64, order 6 for lengths between 65 and 128, and order 8 otherwise. The implementation details are available in src/data/module.py (lines 91–100).
>
> **Decomposition Level**: The decomposition level is based on the sequence length, wavelet family, and wavelet order. Following the PyWavelets framework, we compute the maximum decomposition level as ⌊log₂(sequence_length / (filter_length − 1))⌋. We then clip the resulting value to the range [3, 7] to avoid excessively shallow or deep decompositions, which would result in too few or too many wavelet-level transformer modules, respectively.
>
> We also evaluated custom selections of wavelet order and decomposition levels (Table 5 in the Appendix). The results show that moderate changes to these wavelet configurations lead to similar performance, indicating that WaveletDiff is relatively robust to the specific choice of wavelet parameters.

---

> ### Author Response · Authors · 2026-07-03
>
> 2.
> Note that our initial focus was to only compare WaveletDiff with other diffusion models. Due to our larger model sizes we then compared WaveletDiff with two time series foundation models (TimesFM and Chronos, for which the results were included in the original submission). We believe that these results add to the list of meaningful comparisons, given the substantially larger model sizes of the latter foundational models. Nevertheless, following the reviewer’s recommendations, we additionally included a new baseline, MSDformer (an improved version of SDformer, with both papers now cited in the revised paper), a transformer-based time series generation model. The results show that MSDformer performs better on ETTh1, ETTh2, and particularly on fMRI, whereas WaveletDiff outperforms MSDformer on Exchange Rate and especially on EEG. Furthermore, MSDFormer is a slightly larger model than ours, with larger training times (especially for fMRI data), while both can perform sampling in seconds. The significant performance gaps of the two models on fMRI and EEG, with each model excelling on one of these datasets and outperforming on the other, motivated us to investigate deeper the underlying characteristics of these datasets that distinguish their generative performance under these two models.
>
> | Metric | Methods | ETTh1 | ETTh2 | Stocks | Exchange Rate | fMRI | EEG |
> |:---|:---|:---:|:---:|:---:|:---:|:---:|:---:|
> | Discriminative Score | WaveletDiff | 0.005 | 0.008 | **0.005** | **0.004** | 0.087 | **0.006** |
> |  | MSDformer | **0.004** | **0.006** | 0.011 | 0.008 | **0.008** | 0.053 |
> | Predictive Score | WaveletDiff | **0.119** | **0.106** | **0.037** | 0.037 | 0.100 | **0.000** |
> |  | MSDformer | 0.121 | **0.106** | **0.037** | **0.036** | **0.093** | **0.000** |
> | Context-FID Score | WaveletDiff | 0.020 | 0.023 | 0.018 | **0.006** | 0.104 | **0.006** |
> |  | MSDformer | **0.003** | **0.004** | **0.002** | 0.007 | **0.010** | 0.017 |
> | Correlational Score | WaveletDiff | 0.043 | 0.083 | 0.005 | **0.060** | 1.177 | **1.811** |
> |  | MSDformer | **0.038** | **0.079** | **0.004** | 0.066 | **0.764** | 4.058 |
> | DTW-JS distance | WaveletDiff | 0.101 | **0.064** | 0.106 | **0.121** | 0.191 | **0.055** |
> |  | MSDformer | **0.094** | 0.077 | **0.105** | 0.127 | **0.115** | 0.605 |
> | Training Time (h:m:s) | WaveletDiff | **9:11:06** | **9:00:45** | **4:19:59** | **5:44:06** | **7:52:16** | **8:22:59** |
> |  | MSDformer | 9:36:02 | 9:27:33 | 5:23:36 | 11:47:29 | 25:53:28 | 9:48:10 |
> | Model Parameter (M) | WaveletDiff | **63.1** | **63.1** | 63.1 | **63.1** | **63.2** | **63.1** |
> |  | MSDformer | 72 | 72 | **55.9** | 72 | 83.9 | 72 |
>
> First, among the six datasets, fMRI and EEG represent two extremes in terms of the kurtosis of their wavelet coefficients across decomposition levels. As shown in Figure 3 of the revised main text, EEG exhibits highly non-Gaussian coefficient distributions, whereas the coefficients of fMRI are close to Gaussian. This observation is consistent with prior findings reported in the literature [3, 4] that pointed out that fMRI signals approximately follow a 1/f process, resulting in near-Gaussian wavelet coefficients for each level. The latter explains why WaveletDiff performs relatively poorly on fMRI compared to MSDformer: the near-Gaussian coefficient distributions contain less higher-order statistical structure for our level-specific transformer architecture to exploit. In other words, the near-Gaussian distributions of the coefficients at each level render Gaussian diffusion redundant. On the other hand, for EEG signals, the wavelet coefficients exhibit distributions that exhibit more structure and are individually easier to learn, which makes them highly amenable for wavelet-domain diffusion.
>
> Second, the newly added ablation study in Table 6 in the Appendix, which replaces the level-specific transformers with one single shared transformer, further supports our analysis. Interestingly, the generative performance on fMRI improves on multiple metrics with a shared transformer, whereas the performance of other datasets generally degrades. This supports our hypothesis that level-specific modeling is less beneficial for near-Gaussian wavelet coefficients but more effective for complex coefficient distributions.

---

> ### Author Response · Authors · 2026-07-03
>
> To verify our findings even further, we added new results for time series that share the wavelet characteristics of EEG, such as iEEG and Sleep-EDF data. Again, we observe performance improvements over MSDFormer as for the case of EEG. The results table has also been added to the revised manuscript as Table 8 in the Appendix.
>
> | Metric | Methods | iEEG | Sleep-EDF |
> |---|---|---|---|
> | Discriminative Score | WaveletDiff | **0.009** | **0.004** |
> |  | MSDformer | 0.160 | 0.006 |
> | Predictive Score | WaveletDiff | **0.024** | **0.064** |
> |  | MSDformer | 0.069 | **0.064** |
> | Context-FID Score | WaveletDiff | **0.006** | 0.043 |
> |  | MSDformer | 0.826 | **0.001** |
> | Correlational Score | WaveletDiff | **0.000** | **0.000** |
> |  | MSDformer | **0.000** | **0.000** |
> | DTW-JS distance | WaveletDiff | **0.136** | **0.091** |
> |  | MSDformer | 0.230 | 0.099 |
>
> We also noticed that one evaluation category on which WaveletDiff performs relatively poorly compared to MSDFormer is the Context-FID metric; we believe that this comes from a fundamental architectural difference between the models. WaveletDiff is a diffusion model that directly learns the mapping from a Gaussian prior to the data distribution. In contrast, MSDformer generates samples using a discrete-token transformer operating on the latent manifold learned by a VQ-VAE. Since Context-FID is computed as the FID between TS2Vec embeddings of real and generated sample distributions, the generation process of MSDformer is inherently constrained to remain within the learned low-dimensional latent manifold, resulting in consistently lower Context-FID scores. In contrast, diffusion models learn the data distribution directly from pure Gaussian noise without such manifold constraints. We therefore believe that directly comparing Context-FID between these two fundamentally different generation paradigms may not provide a fully fair assessment of their generation capabilities. Our claim may be further supported by the fact that WaveletDiff underperforms with respect to Context-FID scores when compared to MSDFormer, *it consistently outperforms all other diffusion-based based models on the same*. The above observations warrant future investigations as to why diffusion and MSDFormer models differ so much on this metric.
>
> 3.
> We appreciate the reviewer’s suggestion. However, we were unable to identify a standard or widely adopted benchmark for evaluating generative time series models under noisy or out-of-distribution settings. Also, it may be ill-advised to add so many different diffusion modalities to one paper, especially since we already described - mostly in the Appendix - reproducibility properties of our time-series diffusion model (also, most papers in the diffusion model literature exclusively focus on one of these modalities; for example, ambient diffusion for nontime-series data only examines the influence of noise and nothing else etc). To better demonstrate the robustness of WaveletDiff beyond the clean generation setting, we instead focused on irregularly sampled time series generation.
>
> In the context of irregularly sampled time series generation, we followed the preprocessing protocol of KoVAE and GT-GAN by using cubic spline interpolation to impute missing values, and then trained WaveletDiff on the interpolated time series. As shown in Table 9 in the Appendix, WaveletDiff maintains consistently strong performance across all missing rates.
>
> | Dataset | Missing Rate | Discriminative Score | Predictive Score | Context-FID Score | Correlational Score | DTW-JS Distance |
> |---|---|---|---|---|---|---|
> | ETTh1 | 0 (Regular) | 0.005 | 0.119 | 0.020 | 0.043 | 0.101 |
> |  | 0.3 | 0.014 | 0.120 | 0.019 | 0.064 | 0.115 |
> |  | 0.5 | 0.009 | 0.121 | 0.028 | 0.045 | 0.102 |
> |  | 0.7 | 0.015 | 0.121 | 0.078 | 0.056 | 0.082 |
> | ETTh2 | 0 | 0.008 | 0.106 | 0.023 | 0.083 | 0.064 |
> |  | 0.3 | 0.033 | 0.106 | 0.122 | 0.109 | 0.087 |
> |  | 0.5 | 0.013 | 0.105 | 0.032 | 0.072 | 0.077 |
> |  | 0.7 | 0.016 | 0.108 | 0.049 | 0.073 | 0.095 |
> | Stocks | 0 | 0.005 | 0.037 | 0.018 | 0.005 | 0.106 |
> |  | 0.3 | 0.009 | 0.037 | 0.022 | 0.007 | 0.116 |
> |  | 0.5 | 0.020 | 0.037 | 0.018 | 0.005 | 0.118 |
> |  | 0.7 | 0.011 | 0.037 | 0.027 | 0.006 | 0.131 |
> | Exchange Rate | 0 | 0.004 | 0.037 | 0.006 | 0.060 | 0.121 |
> |  | 0.3 | 0.011 | 0.035 | 0.010 | 0.089 | 0.119 |
> |  | 0.5 | 0.019 | 0.036 | 0.030 | 0.094 | 0.129 |
> |  | 0.7 | 0.010 | 0.036 | 0.010 | 0.067 | 0.136 |
> | fMRI | 0 | 0.087 | 0.100 | 0.104 | 1.177 | 0.191 |
> |  | 0.3 | 0.304 | 0.100 | 0.371 | 1.196 | 0.605 |
> |  | 0.5 | 0.344 | 0.101 | 1.109 | 1.512 | 0.564 |
> |  | 0.7 | 0.380 | 0.102 | 1.407 | 1.781 | 0.318 |
> | EEG | 0 | 0.006 | 0.000 | 0.006 | 1.811 | 0.055 |
> |  | 0.3 | 0.016 | 0.000 | 0.008 | 2.595 | 0.057 |
> |  | 0.5 | 0.010 | 0.000 | 0.010 | 3.683 | 0.054 |
> |  | 0.7 | 0.009 | 0.000 | 0.007 | 4.071 | 0.043 |

---

> ### Author Response · Authors · 2026-07-03
>
> 4.
> We discussed the computational trade-off in Section 4.6 and put the training time and model size of WaveletDiff and MSDformer in Table 1 in the main text. Due to the space limit, we put the inference time in Table 12 in the Appendix.
>
> [1] Stéphane Mallat, A Wavelet Tour of Signal Processing: The Sparse Way.
>
> [2] P. Wojtaszczyk, A Mathematical Introduction to Wavelets.
>
> [3] Ed Bullmore, et al. Wavelets and functional magnetic resonance imaging of the human brain, NeuroImage, 2004.
>
> [4] Voichiţa Maxim, et al. Fractional Gaussian noise, functional MRI and Alzheimer's disease, NeuroImage 2005.

---

### Review · Reviewer_AG1p · 2026-06-21

**Summary Of Contributions:**

The paper proposes a diffusion-based generative model for multivariate time series that operates directly on discrete wavelet coefficients. The technical contributions are: 1) a direct wavelet-domain diffusion framework, 2) cross-scale attention between wavelet decomposition levels, 3) wavelet-aware loss weighting, 4) an optional Parseval-inspired energy regularization term for preserving spectral energy, 5) a DTW-JS metric for alignment-aware distributional comparison. They perform extensive empirical comparisons on six datasets, and an exploratory reproducibility analysis for time-series diffusion models. They compare against FourierDiffusion, Diffusion-TS, TimeGAN, SigDiffusions, KoVAE, and two forecasting foundation models used for generation, where they report strong performance for their proposed method on short sequences of length 24 and longer sequences of lengths 32, 64, and 128, with particularly large improvements in discriminative score and Context-FID.

**Audience:**

Yes

**Audience Explanation:**

The paper is likely to interest a meaningful subset of the TMLR audience, especially researchers working on generative modeling, diffusion models, time-series modeling, representation learning, signal processing, and synthetic data. The application domains are also broad (energy, finance, neuroscience), and other settings where time-series datasets are important but high-quality data may be scarce. The proposed DTW-JS metric and the reproducibility analysis are secondary contributions, but they may also be of interest to readers concerned with evaluating generative time-series models.

**Broader Impact Concerns:**

The paper’s broader impact statement appropriately notes that synthetic time series may not reproduce all characteristics of real data and that use in clinical decision-making, financial risk assessment, or safety-critical engineering can be risky when distributional mismatches are small but consequential.

**Claims And Evidence:**

Yes

**Claims Explanation:**

The evaluation spans multiple domains, several baselines, short and longer sequence lengths, multiple metrics, qualitative visualizations, and meaningful ablations. The results in Table 1 are particularly convincing for the short-sequence setting, where WaveletDiff is usually best or tied across the reported datasets and metrics. Table 2 also supports the claim that the method scales better than several baselines to longer windows. That said, the proposed DTW-JS metric is intuitively motivated, but the paper needs more validation of the metric itself, such as sensitivity to reference-set size, histogram/binning choices, computational cost, and behavior on controlled synthetic examples. Furthermore, WaveletDiff is trained on an H100, while FourierDiffusion baselines are reported on a Tesla T4, so training-time comparisons are not directly interpretable. The paper acknowledges the hardware mismatch, but any claim about efficiency or compute reasonableness should either be moved to a limitation or be supported by same-hardware measurements.

**Requested Changes:**

1. Replace statements implying that WaveletDiff is best on every metric and every sequence length with a more precise claim.

2. The paper uses sliding windows with stride 1, which can create highly overlapping samples. The authors should clearly specify train/validation/test splits, whether overlapping windows can appear across splits, how generated samples are matched to real samples, and how leakage is avoided.

3. Since the best wavelet family is dataset-dependent, the paper should state exactly how the wavelet family/order/level are selected. If choices are tuned per dataset, the tuning protocol should use validation data rather than test metrics.

4. Report baseline parameter counts, training budgets, hyperparameter tuning procedures, and number of random seeds. If some baselines were run with default settings while WaveletDiff was carefully tuned, this should be disclosed.

5. Training-time comparisons across H100 and T4 GPUs should not be used as evidence of relative efficiency. Either rerun key models on the same hardware or present compute results only as descriptive implementation details.

6. Specify the DWT implementation, boundary mode, normalization, and whether the regularizer is theoretically exact or heuristic under each wavelet family.

7. The paper should isolate the contribution of level-specific transformers versus a shared transformer, wavelet-aware loss weighting, and the approximation-level overparameterization.

8. The paper should discuss whether it applies to irregular sampling, missingness, categorical/time-event data, very long sequences, high-dimensional sensor streams, and conditional generation.

---

> ### Author Response · Authors · 2026-07-03
>
> We thank the reviewer for the insightful feedback. Below are our responses; these, in addition to more in-depth results and discussions, have been added to the revised text in blue.
>
> 1.
> We have revised the corresponding statements in the manuscript to make our claims more precise and accurately reflect the experimental results. Specifically, we now state that WaveletDiff outperforms the diffusion-based baselines (FourierDiffusion, Diffusion-TS, and SigDiffusions) on the majority of evaluation metrics, with the smallest margin over FourierDiffusion, while still achieving approximately 3× lower discriminative and Context-FID scores. We also added a detailed comparison with MSDFormer, showing that the two methods achieve largely comparable performance overall, with the main differences occurring on the fMRI and EEG datasets, where WaveletDiff trails on fMRI but outperforms MSDFormer on EEG. Finally, we included an analysis of the wavelet coefficient distributions of the fMRI and EEG datasets, demonstrating that fMRI exhibits an approximately Gaussian coefficient distribution, whereas EEG has a highly non-Gaussian distribution, which helps explain the observed performance differences.
>
> 2.
> For training the generative model, we followed the previous standard protocol using all dataset with stride 1 as training data (no val and test set). The train/test split is only used for the Discriminative score evaluation, where we followed TimeGAN’s setting using 80/20 train/test split, and the leakage is avoided by dividing the two sets before cropping into windows.
>
> 3.
> **Wavelet Family**: The default wavelet family in WaveletDiff is the Daubechies wavelet family, which achieves consistently strong performance across all datasets (see Table 4 in the Appendix). The ability to select different wavelet families for different signals is an additional feature of WaveletDiff that can lead to further improvements in generative performance on specific datasets. In our experiments, dataset-specific wavelet selection was performed by choosing the wavelet family with the best validation performance, which requires additional hyperparameter tuning and computational cost. However, to point out once again, the results in Table 4 demonstrate that such tuning is not really necessary in order to obtain strong performance in general, as the default Daubechies wavelet family already performs competitively across all benchmarks. Furthermore, it is important to notice that rigorous mathematical results are readily available to suggest which wavelet families are most amenable for which types of time series data decomposition/analysis (i.e., stationarity/nonstationarity, smoothness, sparseness etc). We added a citation to these works [1, 2] but strongly emphasize the fact that these results pertain to signal analysis, and not signal generation. Analysis and generation are very different tasks and often, as we saw in our comparative studies, what makes a data class amenable for analysis hurts generation and vice versa (e.g., fMRI data).
>
> **Wavelet Order**: The wavelet order is selected based on the sequence length to avoid excessive boundary artifacts and ensure a meaningful multilevel decomposition. Specifically, for Daubechies wavelets, we use order 2 for sequence lengths ≤ 32, order 4 for lengths between 33 and 64, order 6 for lengths between 65 and 128, and order 8 otherwise. The implementation details are available in src/data/module.py (lines 91–100).
>
> **Decomposition Level**: The decomposition level is based on the sequence length, wavelet family, and wavelet order. Following the PyWavelets framework, we compute the maximum decomposition level as ⌊log₂(sequence_length / (filter_length − 1))⌋. We then clip the resulting value to the range [3, 7] to avoid excessively shallow or deep decompositions, which would result in too few or too many wavelet-level transformer modules, respectively.
>
> We also evaluated custom selections of wavelet order and decomposition levels (Table 5 in the Appendix). The results show that moderate changes to these wavelet configurations lead to similar performance, indicating that WaveletDiff is relatively robust to the specific choice of wavelet parameters.

---

> ### Author Response · Authors · 2026-07-03
>
> 4.
> All baseline models were trained using the default hyperparameter settings and configurations provided in their official implementations. We did not perform random seed selection for either WaveletDiff or the baseline methods. For WaveletDiff, we tuned standard hyperparameters that are commonly optimized for diffusion Transformer models, including the Transformer embedding dimension, number of layers, learning rate, diffusion noise schedule, number of diffusion steps, and training epochs. These hyperparameters are not specific to WaveletDiff and would similarly require tuning for other diffusion-based baselines. The only WaveletDiff-specific hyperparameters are the wavelet configuration and the energy regularization weight. As discussed in our response to Question 3, we found that selecting a Daubechies wavelet with an appropriate, yet simple to determine decomposition level based on the sequence length provides consistently good performance. The energy regularization weight is the only hyperparameter unique to WaveletDiff that requires tuning, and it is only used for the long-sequence generation tasks. We note that, for each hyperparameter, we explored only a handful of candidate values, resulting in a relatively modest tuning effort rather than an exhaustive hyperparameter search.
>
> 5.
> We agree that training times obtained on different GPU hardware should not be used to draw conclusions regarding the relative computational efficiency of different methods. Our intention was not to present the runtime measurements as an efficiency comparison between WaveletDiff and FourierDiffusion. Instead, the reported training times were included as implementation details to provide readers with a practical understanding of the computational resources required by each model under their respective experimental settings. To avoid potential ambiguity, we added the training time comparison between WaveletDiff and MSDformer on the same device (NVIDIA A100), which allows for a completely fair comparison. Table 1 shows that our model requires less training time, particularly on fMRI, where it finishes training in only one-third of  the time required by MSDformer.
>
> 6.
> We use the PyWavelets implementation for all DWT and IDWT operations. Boundary effects are handled using symmetric extension, as described in Section 3.1. Regarding normalization, we follow the standard normalization conventions of the selected wavelet family as implemented in PyWavelets. For the energy regularization, since we are using the symmetric extension, the regularizer is heuristic. We have added these details in the revised manuscript.
>
> 7.
> The wavelet-aware loss weighting ablation study is already provided in Table 7 in the Appendix. We have also added the two new ablation studies (shared transformer and w/o larger approx.-level capacity) in Table 6 in the revised manuscript.

---

> ### Author Response · Authors · 2026-07-03
>
> 8.
> We appreciate the reviewer’s suggestion. In the revised manuscript, we have added one more evaluation task: irregularly sampled time series generation (the relationship with forecasting was studied in the “opposite” direction in our initially submitted paper, and we would really like to focus on generative tasks, and not diverge into forecasting). Although we agree that extending WaveletDiff to other settings such as contextual generation, categorical/event data and high-dimensional sensor streams would be valuable future directions, we think that adding them to the paper would be ill advised. Our goal is not to cover as many possible modifications of wavelet diffusion but rather focus on a deeper understanding why and how it works. We also had results pertaining to reproducibility included in the paper - adding more directions would “water down” the main message (note that most papers in the diffusion model literature exclusively focus on one of all these modalities; for example, ambient diffusion for nontime-series data only examines the influence of noise and nothing else etc). We believe the newly added experiments already provide substantially stronger evidence of WaveletDiff’s robustness and applicability beyond unconditional generation on clean benchmark datasets.
>
> In the context of irregularly sampled time series generation, we followed the preprocessing protocol of KoVAE and GT-GAN by using cubic spline interpolation to impute missing values, and then trained WaveletDiff on the interpolated time series. As shown in Table 9 in the Appendix, WaveletDiff maintains consistently strong performance across all missing rates.
>
> | Dataset | Missing Rate | Discriminative Score | Predictive Score | Context-FID Score | Correlational Score | DTW-JS Distance |
> |---|---|---|---|---|---|---|
> | ETTh1 | 0 (Regular) | 0.005 | 0.119 | 0.020 | 0.043 | 0.101 |
> |  | 0.3 | 0.014 | 0.120 | 0.019 | 0.064 | 0.115 |
> |  | 0.5 | 0.009 | 0.121 | 0.028 | 0.045 | 0.102 |
> |  | 0.7 | 0.015 | 0.121 | 0.078 | 0.056 | 0.082 |
> | ETTh2 | 0 | 0.008 | 0.106 | 0.023 | 0.083 | 0.064 |
> |  | 0.3 | 0.033 | 0.106 | 0.122 | 0.109 | 0.087 |
> |  | 0.5 | 0.013 | 0.105 | 0.032 | 0.072 | 0.077 |
> |  | 0.7 | 0.016 | 0.108 | 0.049 | 0.073 | 0.095 |
> | Stocks | 0 | 0.005 | 0.037 | 0.018 | 0.005 | 0.106 |
> |  | 0.3 | 0.009 | 0.037 | 0.022 | 0.007 | 0.116 |
> |  | 0.5 | 0.020 | 0.037 | 0.018 | 0.005 | 0.118 |
> |  | 0.7 | 0.011 | 0.037 | 0.027 | 0.006 | 0.131 |
> | Exchange Rate | 0 | 0.004 | 0.037 | 0.006 | 0.060 | 0.121 |
> |  | 0.3 | 0.011 | 0.035 | 0.010 | 0.089 | 0.119 |
> |  | 0.5 | 0.019 | 0.036 | 0.030 | 0.094 | 0.129 |
> |  | 0.7 | 0.010 | 0.036 | 0.010 | 0.067 | 0.136 |
> | fMRI | 0 | 0.087 | 0.100 | 0.104 | 1.177 | 0.191 |
> |  | 0.3 | 0.304 | 0.100 | 0.371 | 1.196 | 0.605 |
> |  | 0.5 | 0.344 | 0.101 | 1.109 | 1.512 | 0.564 |
> |  | 0.7 | 0.380 | 0.102 | 1.407 | 1.781 | 0.318 |
> | EEG | 0 | 0.006 | 0.000 | 0.006 | 1.811 | 0.055 |
> |  | 0.3 | 0.016 | 0.000 | 0.008 | 2.595 | 0.057 |
> |  | 0.5 | 0.010 | 0.000 | 0.010 | 3.683 | 0.054 |
> |  | 0.7 | 0.009 | 0.000 | 0.007 | 4.071 | 0.043 |
>
> 9.
> Following the reviewer's suggestion, we conducted a series of analyses to further validate the utility of the proposed DTW-JS distance metric.
>
> **Sensitivity to reference-set size.** We randomly sampled 100 sequences each from the generated and real datasets, and drew the reference set M from their union. Using the EEG data as a representative example, we varied the reference-set size from 5 to 200 samples. As shown in Figure 4 in the Appendix, DTW-JS is unstable for small reference sets (n<50) but converges and remains stable for larger sets (n>75). In practice we use a reference set of 100 samples throughout our experiments.
>
> **Sensitivity to histogram/binning choice.** Fixing the reference set at 100 samples, we analyzed the sensitivity of DTW-JS to the number of bins used to form the distance distributions, with equal-width binning. As shown in Figure 5 in the Appendix, too few bins (<30) make the distributions overly coarse and systematically underestimate the DTW-JS distance, whereas too many bins (>100) make them overly fine and inflate the distance. We use 50 bins throughout, which yields a stable estimate, and apply the same setting when evaluating all baseline models to ensure a fair comparison.
>
> **Computational cost.** Under the default setting, with the real, generated, and reference sets each containing 100 samples, computing the DTW-JS distance once takes 5.77 seconds on average. Following the same protocol as the other four metrics, we repeat the computation five times and report the mean and standard deviation, giving a total cost of roughly 30 seconds per evaluation. Behavior on controlled synthetic examples are still running and will be reported shortly.
>
> [1] Stéphane Mallat, A Wavelet Tour of Signal Processing: The Sparse Way
>
> [2] P. Wojtaszczyk, A Mathematical Introduction to Wavelets.

---

### Review · Reviewer_GhMn · 2026-06-22

**Summary Of Contributions:**

The main contribution of WaveletDiff is a wavelet-domain diffusion framework for time series generation, which explicitly models multi-resolution structures via level-specific Transformers, enables cross-scale information exchange through adaptive cross-level attention, and preserves time-frequency energy properties using Parseval-based constraints. Extensive experiments on six real-world datasets demonstrate the good performance of WaveletDiff.

Strengths:

1. The structure is easy to follow.
2. The paper is well-structured with comprehensive experiments.
3. The experimental results appear to be reproducible.

Weaknesses:

1. The novelty of the paper is limited. Multi-scale modeling and wavelet transforms are already well-established techniques in time series analysis, and their integration into a diffusion-based generative framework appears to be a relatively incremental extension rather than a fundamentally new contribution.

2. The discussion on ”the choice of the wavelet domain“at the end of Section 3 lacks sufficient theoretical support. A more rigorous justification would strengthen the paper, for instance by analyzing the effectiveness of the wavelet transformation from the perspective of information preservation in the transformed frequency-domain representations.

3. The experimental evaluation would be strengthened by including more competitive generative model-based baselines. For example, recent methods such as KoVAE [1], FlowTS [2], and SDformer [3] are not considered in the current comparison. Adding these baselines would provide a more convincing and comprehensive evaluation of the advantages of WaveletDiff.

4. The motivation for introducing the newly proposed evaluation metric is not sufficiently clear. As currently described, both DTW-distance and Context-FID are used to evaluate the discrepancy between the generated and real distributions. If this interpretation is correct, it remains unclear why DTW-JS distance is additionally needed. The authors should clarify the unique role of DTW-JS distance and explain what complementary information it provides beyond the existing metrics.

[1]. Naiman, Ilan, et al. "Generative modeling of regular and irregular time series data via Koopman VAEs." International conference on learning representations. Vol. 2024. 2024.

[2]. Hu, Yang, et al. "FlowTS: Time Series Generation via Rectified Flow." arXiv preprint arXiv:2411.07506 (2024).

[3]. Chen, Zhicheng, et al. "Sdformer: Similarity-driven discrete transformer for time series generation." Advances in Neural Information Processing Systems 37 (2024): 132179-132207.

**Audience:**

Yes

**Audience Explanation:**

Yes. The paper addresses an important problem in time series generation and presents empirical findings that may be of interest to researchers working on generative models, multi-scale time series modeling, and diffusion-based methods. However, the current contribution appears somewhat incremental, and the broader interest would be strengthened by clearer theoretical justification and more competitive comparisons.

**Broader Impact Concerns:**

I do not identify any major ethical concerns.

**Claims And Evidence:**

Yes

**Claims Explanation:**

1. The paper claims the effectiveness of performing diffusion modeling in the wavelet domain; however, this claim lacks sufficient theoretical support. A more rigorous analysis is needed to justify why the wavelet domain is preferable to the time domain or other transformation domains for time series generation. Without such justification, the advantage of wavelet-domain diffusion remains mainly empirical rather than theoretically grounded.

2. The task settings in Sections 4.2 and 4.3 are not clearly described. Time series generation generally includes two types of settings: unconditional generation and conditional generation. However, the paper does not explicitly clarify which type of generation is considered in these sections.

**Requested Changes:**

Please refer to the weaknesses listed above. I consider the missing theoretical justification for the wavelet-domain diffusion design, the unclear motivation for the proposed evaluation metric, and the lack of competitive generative baselines. Addressing the unclear task descriptions and improving the discussion would further strengthen the paper.

---

> ### Author Response · Authors · 2026-07-03
>
> We thank the reviewer for the insightful feedback. Below are our responses; these, in addition to more in-depth results and discussions, have been added to the revised text in blue.
>
> 1.
> We agree that wavelet transforms and multi-scale modeling are well-established techniques in time series analysis, but not in the domain of time series generation, which is a fundamentally different task. Our overarching goal was to address the question of how wavelet representations can be effectively incorporated into diffusion-based generative modeling and which signals are amenable for wavelet domain diffusion. Note that the use of wavelets in generative settings raises many challenges, such as modeling dependencies across decomposition levels and designing appropriate denoising strategies in the wavelet domain. WaveletDiff introduces several design choices, including per-level processing and cross-level attention, to address these challenges. We therefore view the contribution as a principled adaptation of wavelet representations to diffusion-based generation rather than a simple combination of existing techniques. As an example, we would like to point out the case of the highly different performance of WaveletDiff on fMRI and EEG data - see our detailed explanation under bullet 3. Finally, we note that TMLR submissions should not be judged solely on the basis of novelty.
>
> 2.
> With regards to the reviewer’s question regarding the technical support for our wavelet choices: In the initial submission, we only discussed *which wavelets are amenable for which signal types for analysis purposes*. Here, we are trying to exploit wavelet decompositions for *generative purposes*, and generation is a very different task. Combining the rather involved statistical analyses of wavelet decomposition properties with diffusion process analysis is a very difficult task and would warrant a completely new paper submission.
>
> 3.
> We note that KoVAE is already included as one of the baselines in our original submission. We also investigated FlowTS, but its public repository currently does not provide the code required to reproduce the main experimental results, preventing us from performing a fair comparison during the rebuttal period. Following the reviewer’s suggestion, we have additionally included MSDformer (an improved version of SDformer, with both papers now cited in the revised paper), a transformer-based time series generation model. The results show that MSDformer performs better on ETTh1, ETTh2, and particularly on fMRI, whereas WaveletDiff outperforms MSDformer on Exchange Rate and especially on EEG. Furthermore, MSDFormer is a slightly larger model than ours, with larger training times (especially for fMRI data), while both can perform sampling in seconds. The significant performance gaps of the two models on fMRI and EEG, with each model excelling on one of these datasets and outperforming on the other, motivated us to investigate deeper the underlying characteristics of these datasets that distinguish their generative performance under these two models.
>
> | Metric | Methods | ETTh1 | ETTh2 | Stocks | Exchange Rate | fMRI | EEG |
> |:---|:---|:---:|:---:|:---:|:---:|:---:|:---:|
> | Discriminative Score | WaveletDiff | 0.005 | 0.008 | **0.005** | **0.004** | 0.087 | **0.006** |
> |  | MSDformer | **0.004** | **0.006** | 0.011 | 0.008 | **0.008** | 0.053 |
> | Predictive Score | WaveletDiff | **0.119** | **0.106** | **0.037** | 0.037 | 0.100 | **0.000** |
> |  | MSDformer | 0.121 | **0.106** | **0.037** | **0.036** | **0.093** | **0.000** |
> | Context-FID Score | WaveletDiff | 0.020 | 0.023 | 0.018 | **0.006** | 0.104 | **0.006** |
> |  | MSDformer | **0.003** | **0.004** | **0.002** | 0.007 | **0.010** | 0.017 |
> | Correlational Score | WaveletDiff | 0.043 | 0.083 | 0.005 | **0.060** | 1.177 | **1.811** |
> |  | MSDformer | **0.038** | **0.079** | **0.004** | 0.066 | **0.764** | 4.058 |
> | DTW-JS distance | WaveletDiff | 0.101 | **0.064** | 0.106 | **0.121** | 0.191 | **0.055** |
> |  | MSDformer | **0.094** | 0.077 | **0.105** | 0.127 | **0.115** | 0.605 |
> | Training Time (h:m:s) | WaveletDiff | **9:11:06** | **9:00:45** | **4:19:59** | **5:44:06** | **7:52:16** | **8:22:59** |
> |  | MSDformer | 9:36:02 | 9:27:33 | 5:23:36 | 11:47:29 | 25:53:28 | 9:48:10 |
> | Model Parameter (M) | WaveletDiff | **63.1** | **63.1** | 63.1 | **63.1** | **63.2** | **63.1** |
> |  | MSDformer | 72 | 72 | **55.9** | 72 | 83.9 | 72 |

---

> ### Author Response · Authors · 2026-07-03
>
> First, among the six datasets, fMRI and EEG represent two extremes in terms of the kurtosis of their wavelet coefficients across decomposition levels. As shown in Figure 3 of the revised main text, EEG exhibits highly non-Gaussian coefficient distributions, whereas the coefficients of fMRI are close to Gaussian. This observation is consistent with prior findings reported in the literature [1, 2] that pointed out that fMRI signals approximately follow a 1/f process, resulting in near-Gaussian wavelet coefficients for each level. The latter explains why WaveletDiff performs relatively poorly on fMRI compared to MSDformer: the near-Gaussian coefficient distributions contain less higher-order statistical structure for our level-specific transformer architecture to exploit. In other words, the near-Gaussian distributions of the coefficients at each level render Gaussian diffusion redundant. On the other hand, for EEG signals, the wavelet coefficients exhibit distributions that exhibit more structure and are individually easier to learn, which makes them highly amenable for wavelet-domain diffusion.
>
> Second, the newly added ablation study in Table 6 in the Appendix, which replaces the level-specific transformers with one single shared transformer, further supports our analysis. Interestingly, the generative performance on fMRI improves on multiple metrics with a shared transformer, whereas the performance of other datasets generally degrades. This supports our hypothesis that level-specific modeling is less beneficial for near-Gaussian wavelet coefficients but more effective for complex coefficient distributions.
>
> To verify our findings even further, we added new results for time series that share the wavelet characteristics of EEG, such as iEEG and Sleep-EDF data. Again, we observe performance improvements over MSDFormer as for the case of EEG. The results table has also been added to the revised manuscript as Table 8 in the Appendix.
>
> | Metric | Methods | iEEG | Sleep-EDF |
> |---|---|---|---|
> | Discriminative Score | WaveletDiff | **0.009** | **0.004** |
> |  | MSDformer | 0.160 | 0.006 |
> | Predictive Score | WaveletDiff | **0.024** | **0.064** |
> |  | MSDformer | 0.069 | **0.064** |
> | Context-FID Score | WaveletDiff | **0.006** | 0.043 |
> |  | MSDformer | 0.826 | **0.001** |
> | Correlational Score | WaveletDiff | **0.000** | **0.000** |
> |  | MSDformer | **0.000** | **0.000** |
> | DTW-JS distance | WaveletDiff | **0.136** | **0.091** |
> |  | MSDformer | 0.230 | 0.099 |
>
> We also noticed that one evaluation category on which WaveletDiff performs relatively poorly compared to MSDFormer is the Context-FID metric; we believe that this comes from a  fundamental architectural difference between the models. WaveletDiff is a diffusion model that directly learns the mapping from a Gaussian prior to the data distribution. In contrast, MSDformer generates samples using a discrete-token transformer operating on the latent manifold learned by a VQ-VAE. Since Context-FID is computed as the FID between TS2Vec embeddings of real and generated sample distributions, the generation process of MSDformer is inherently constrained to remain within the learned low-dimensional latent manifold, resulting in consistently lower Context-FID scores. In contrast, diffusion models learn the data distribution directly from pure Gaussian noise without such manifold constraints. We therefore believe that directly comparing Context-FID between these two fundamentally different generation paradigms may not provide a fully fair assessment of their generation capabilities. Our claim may be further supported by the fact that WaveletDiff underperforms with respect to Context-FID scores when compared to MSDFormer, *it consistently outperforms all other diffusion-based based models on the same*. The above observations warrant future investigations as to why diffusion and MSDFormer models differ so much on this metric.

---

> ### Author Response · Authors · 2026-07-03
>
> 4.
> The motivation for proposing DTW-JS distance is described in Section 4.1. Context-FID quantifies distributional distance in the embedding space of a pretrained TS2Vec encoder, and therefore inherits whatever structural biases that encoder may have. In particular, it is unclear whether the encoder adequately captures temporal alignment, phase shifts, or nonlinear time distortions. Dynamic Time Warping, on the other hand, is a model-free, alignment-aware similarity measure that explicitly accounts for temporal warping by finding the optimal monotone alignment between two sequences. This makes DTW sensitive to shape-based similarity in a way that embedding-based distances may not be. Since raw DTW distances are between individual sequence pairs, we therefore extend it to distributional distance using Jensen–Shannon divergence. We have also expanded the discussion of this motivation in Section 4.1 of the revised manuscript.
>
> Additional notes: We have explicitly specified that Sections 4.2 and 4.3 focus on “unconditional” time series generation in the revised manuscript.
>
> [1] Ed Bullmore, et al. Wavelets and functional magnetic resonance imaging of the human brain, NeuroImage, 2004.
>
> [2] Voichiţa Maxim, et al. Fractional Gaussian noise, functional MRI and Alzheimer's disease, NeuroImage 2005.